# R1-ShareVL: Incentivizing Reasoning Capability of Multimodal Large Language Models via Share-GRPO

**Huanjin Yao**[2,3*]**, Qixiang Yin**[4*]**, Jingyi Zhang**[1]**, Min Yang**[2]**, Yibo Wang**[3]**, Wenhao Wu**[5]
**Fei Su**[4]**, Li Shen**[6]**, Minghui Qiu**[2]**, Dacheng Tao**[1✉]**, Jiaxing Huang**[1✉]

[1]Nanyang Technological University    [2]ByteDance    [3]Tsinghua University
[4]Beijing University of Posts and Telecommunications
[5]The University of Sydney    [6] Sun Yat-sen University
[*] Equal Contribution    [✉] Corresponding Author

## Abstract

In this work, we aim to incentivize the reasoning ability of Multimodal Large Language Models (MLLMs) via reinforcement learning (RL) and develop an effective approach that mitigates the sparse reward and advantage vanishing issues during RL. To this end, we propose Share-GRPO, a novel RL approach that tackle these issues by exploring and sharing diverse reasoning trajectories over expanded question space. Specifically, Share-GRPO first expands the question space for a given question via data transformation techniques, and then encourages MLLM to effectively explore diverse reasoning trajectories over the expanded question space and shares the discovered reasoning trajectories across the expanded questions during RL. In addition, Share-GRPO also shares reward information during advantage computation, which estimates solution advantages hierarchically across and within question variants, allowing more accurate estimation of relative advantages and improving the stability of policy training. Extensive evaluations over 6 widely-used reasoning benchmarks showcase the superior performance of our method. Code is available at `https://github.com/HJYao00/R1-ShareVL`.

## 1 Introduction

The recent success of Reinforcement Learning (RL) in Large Language Models (LLMs), such as Kimi-K1.5 [1] and DeepSeek-R1 [2], shows its promise in incentivizing model's long-chain reasoning capability, enabling LLMs to tackle complex tasks such as mathematical and scientific reasoning. The core design of these advances (*e.g.*, GRPO [3] in Deepseek-R1) lies in online reinforcement learning without the need of reward models, which encourages an LLM to generate a group of reasoning paths and iteratively refine its reasoning process with a group relative advantage estimation mechanism based on rule-based reward functions. Typically, a simple reward strategy is adopted: reasoning paths leading to correct answers receive higher rewards, while those leading to incorrect answers receive lower ones, where the model is optimized via the group relative advantages estimated from the rewards.

Inspired by these advancements, we aim to develop a simple and effective reinforcement learning method for Multimodal LLMs (MLLMs) to incentivize their long-chain reasoning ability. A simple way is to directly apply these LLM online reinforcement learning methods like GRPO on MLLMs. However, we empirically observe that directly applying GRPO on MLLMs suffers from sparse reward and advantage vanishing issues, leading to degraded performance in enhancing MLLM's reasoning capability [4, 5, 6]:

39th Conference on Neural Information Processing Systems (NeurIPS 2025).

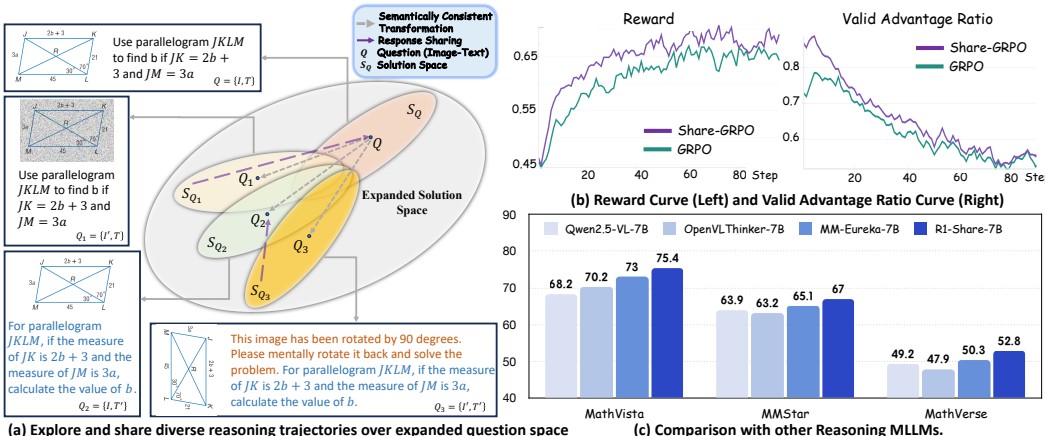

Figure 1: **(a)** Share-GRPO expands the question space via semantically consistent transformations, and then explores diverse reasoning trajectories from different question variants and shares the discovered trajectories among them. **(b)** Share-GRPO provides denser rewards and higher valid advantage ratios compared to GRPO, demonstrating its effectiveness in mitigating sparse reward and advantage vanishing issues. **(c)** Share-GRPO outperforms the baseline and other SOTA RL-based reasoning MLLMs on both mathematical and general reasoning benchmarks.

**(1) Sparse reward:** Most current MLLMs, especially smaller ones, exhibit very limited long-chain reasoning capability. As a result, only a few generated reasoning paths receive positive rewards, especially on challenging questions and particularly during the early stage of training. This leads to sparse rewarding, inefficient exploration and instable training in GRPO-like methods.

**(2) Advantage vanishing:** GRPO-like methods compute relative advantages by comparing the rewards of a group of responses sampled from a given question, leading to advantage vanishing when receiving homogeneous responses. Specifically, along reinforcement learning process, the model tends to gradually predict similar and all correct responses for well-learned questions, and similar and all incorrect responses for poor-learned questions. In this way, the relative advantages tend to approach zero when the group of responses become more homogeneous, and collapse to zero when all responses receive identical rewards (*e.g.*, all correct or all incorrect), resulting ineffective reinforcement learning.

Motivated by these observations, we propose Share-GRPO, a novel approach that introduces the concept of information sharing into MLLM reinforcement learning to mitigate sparse reward and advantage vanishing issues. The core idea of Share-GRPO lies in exploring and sharing diverse reasoning trajectories over expanded question space as shown in Fig. 1 (a). Specifically, Share-GRPO first expands the question space for a given question via data transformation techniques, and then encourages MLLM to effectively explore diverse reasoning trajectories over the expanded question space and shares the discovered reasoning trajectories across the expanded questions during reinforcement learning. In this way, each expanded question variant can both contribute and benefit from the reasoning trajectories generated by others in the expand question space, allowing the model to jointly explore and learn from a shared solution space across expanded questions.

In addition, Share-GRPO also shares reward information during advantage computation, which estimates solution advantages hierarchically across and within question variant, allowing more accurate estimation of relative advantages and improving the stability of policy training. Specifically, we estimate advantages at two levels: a local level, which consists of responses generated from each individual question variant, and a global level, which aggregates responses across all variants of the same seed question. This hierarchical advantage estimation enables more robust and fine-grained relative advantage computation, where the local level captures intra-variant structure and variance while the global level exploits cross-variant diversity and complementarity and stabilizes reward signals.

In this way, Share-GRPO effectively mitigates the sparse reward and advantage vanishing issues: (1) Share-GRPO expands the question space and enables more diverse solution space for each given question, which effectively increases the likelihood of generating a successful reasoning response and

thus mitigates the sparse rewarding issue as illustrated in the left curve of Fig. 1 (b). (2) Share-GRPO allows the model to explore diverse reasoning trajectories from the expanded question space and shares the discovered reasoning trajectories, ultimately mitigating the advantage vanishing issue effectively as illustrated in the right curve in Fig. 1 (b). (3) Share-GRPO estimates solution advantages hierarchically across and within question variant, which enables more accurate estimation of relative advantages and stable reinforcement learning process.

In summary, the main contributions of this work are summarized as follows: First, we introduce the concept of information sharing into MLLM reinforcement learning, and propose Share-GRPO which explores and shares diverse reasoning trajectories over expanded question space, effectively mitigating the sparse reward and advantage vanishing issues. To the best of our knowledge, this is the first work that explores information sharing for MLLM reasoning reinforcement learning. Second, we design a hierarchical advantage estimation method by sharing reward information, which estimates solution advantages hierarchically across and within question variant, allowing accurate and robust advantage estimation. Third, extensive experiments on 6 MLLM reasoning benchmarks demonstrate the superiority of our proposed methods as illustrated in Fig. 1 (c).

## 2 Related Work

### 2.1 Multimodal Large Language Model

Multimodal Large Language Models (MLLMs) [7, 8, 9, 10, 11, 12, 13, 14, 15, 16, 17, 18] demonstrate outstanding performance in semantic understanding of cross-domain visual content and multimodal reasoning. Early research on MLLMs primarily focused on text-image alignment and the integration of multiple modalities [19, 20, 21, 22, 23]. Subsequently, models like GPT-4V [24] achieved breakthroughs in cross-modal understanding through multimodal instruction fine-tuning, enabling them to support simple tasks such as image captioning, visual question answering and OCR. More complex tasks, such as mathematical reasoning, document understanding, etc., require MLLMs to be able to perform complex logical deductions. For MLLM reasoning, models such as Multimodal-CoT [25] and LLaVA-CoT [26] employ chain-of-thought (CoT) reasoning, breaking down the multimodal reasoning process into step-by-step inference steps while leveraging multimodal data to improve the model's reasoning capabilities. Additionally, Mulberry [27] proposes CoMCTS to generate effective reasoning paths through multi-model collaboration. Different from these studies, this work focuses on reinforcement learning to improve MLLM reasoning capability.

### 2.2 Reinforcement Learning for Multimodal Large Language Model Reasoning

Reinforcement learning has become an essential technology for enhancing the capabilities of MLLMs. Early research primarily focused on Reinforcement Learning from Human Feedback(RLHF) [28, 29, 30, 31], which aligns the outputs of multimodal models with human preferences by incorporating human feedback signals. Recently, DeepSeek-R1 [2] utilizes a simple rule-based reward function to provide effective and reliable reward signals during the RL process. This indicates that the Group Relative Policy Optimization (GRPO) with result-level rewards effectively enhances the reasoning ability of LLMs [32, 2, 33]. In the multimodal domain, researchers have begun exploring the use of RL to enhance the visual reasoning capabilities of MLLMs. Recent works, such as Vision-R1 [34] and MM-Eureka [5] have open-sourced large-scale SFT cold start data and RL data. R1-V [35], Reason-RFT [36], R1-VL [4] and other methods [37, 38, 39, 40] have designed various rule-based reward functions to enhance the reasoning abilities of MLLMs, such as geometric understanding and spatial perception. Unlike these methods, our Share-GRPO explores information sharing for MLLM reasoning reinforcement learning to mitigate sparse reward and advantage vanishing issues.

### 2.3 Information Sharing in Deep Learning

Information sharing is a key strategy in deep learning, enabling more effective learning through the exchange of signals across modalities, tasks, or hierarchical model components. In multi-modal learning, models such as ViLBERT [41] and LXMERT [42] employ cross-modal attention to achieve fine-grained information fusion between vision and language streams. In contrastive learning (*e.g.*, SimCLR [43], MoCo [44]), shared representations across augmented views enhance feature robustness. This concept extends to reinforcement learning, especially in multi-task and

multi-agent settings, where information sharing improves sample efficiency and mitigates sparse rewards. Methods like Distral [45] and PopArt [46] promote shared policy structures, while agents in multi-agent RL benefit from shared value functions or communication protocols [47, 48]. [49] further demonstrate that shared representations enhance generalization in multi-task RL. Unlike prior work, we introduce information sharing into MLLM reasoning reinforcement learning to mitigate sparse rewards and advantage vanishing for more effective reasoning learning.

# 3 Method

This section first provides the preliminary of Group Relative Policy Optimization (GRPO), and then presents the proposed Share-GRPO that introduces the concept of information sharing into MLLM reinforcement learning. Further details are elaborated in the subsequent subsections.

## 3.1 Preliminary

**Group Relative Policy Optimization (GRPO)**. GRPO [3] is a variant of Proximal Policy Optimization (PPO) [50], designed to enhance the performance of LLMs on complex reasoning tasks, such as mathematical and scientific reasoning. Starting with a pretrained MLLM to be optimized, GRPO first uses it to initialize a policy model $\pi_\theta$ and a reference model $\pi_{\text{old}}$. For a given image-text pair $(I, T)$, the reference policy model $\pi_{\theta_{\text{old}}}$ generates a set of responses $\{o_1, o_2, ..., o_G\}$. A group-based reward function then computes the corresponding rewards $\{R_1, R_2, ..., R_G\}$, which are subsequently used to estimate the advantage $\hat{A}i$ for each response relative to the group:

$$\hat{A}_i = \frac{R_i - \text{mean}\left(\{R_i\}_{i=1}^G\right)}{\text{std}\left(\{R_i\}_{i=1}^G\right)}.$$ (1)

Similar to PPO, GRPO employs a clipped objective with a KL penalty term:

$$\mathcal{J}_{\text{GRPO}}(\theta) = \mathbb{E}_{(I,T)\sim p_\mathcal{D}, o\sim\pi_{\theta_{\text{old}}}(\cdot|I,T)}$$
$$\left[\frac{1}{n}\sum_{i=1}^n \min\left(\frac{\pi_\theta(o_i \mid I, T)}{\pi_{\theta_{\text{old}}}(o_i \mid I, T)}\hat{A}_i, \text{clip}\left(\frac{\pi_\theta(o_i \mid I, T)}{\pi_{\theta_{\text{old}}}(o_i \mid I, T)}, 1-\epsilon, 1+\epsilon\right)\hat{A}_i - \beta D_{\text{KL}}\left(\pi_\theta || \pi_{\text{ref}}\right)\right)\right].$$ (2)

**Sparse Reward and Advantage Vanishing Issues.** Despite the effectiveness of GRPO, it generally faces two challenges when applied to MLLMs: the sparse reward issue and the advantage vanishing issue. Sparse rewarding arises due to the limited reasoning ability of current MLLMs, where only a few reasoning paths receive positive rewards, leading to inefficient exploration and instable training. To alleviate this, prior work such as R1-VL [4] introduces step-wise reward signals to provide dense rewards throughout the reasoning process. Advantage vanishing occurs when MLLMs generate homogeneous responses for the same question and receive identical rewards, causing the relative advantages to collapse to zero and resulting in ineffective reinforcement learning. To tackle this issue, VL-Rethinker [6] and Skywork R1 [51] select the samples with large magnitudes of advantages and reuse them in RL process, while MM-Eureka [5] employs an online filtering strategy to remove the samples with zero advantage. Different from the prior works, our Share-GRPO effectively addresses both of these two challenges by exploring and sharing diverse reasoning trajectories over expanded question space, therefore encouraging reward diversity and stable policy optimization.

## 3.2 Share-GRPO

We propose Share-GRPO, a novel online MLLM reinforcement learning framework that mitigates the sparse reward and advantage vanishing issues via exploring and sharing diverse reasoning trajectories over expanded question space. Specifically, for a given question, Share-GRPO first applies semantically consistent transformation to generate a set of varied but semantically equivalent questions, thereby expanding the question space. It then encourages the MLLM to explore diverse reasoning paths over the expanded question space and facilitates the sharing of discovered reasoning trajectories and their rewards across the expanded questions during the reinforcement learning process, as illustrated in Fig. 2.

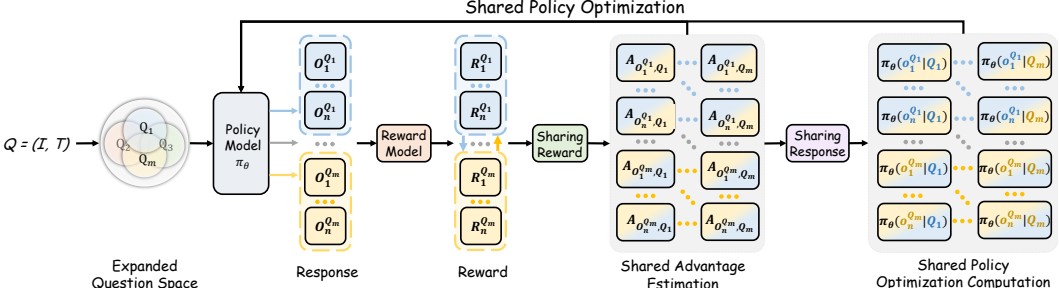

Figure 2: Overview of the proposed Share-GRPO. For a given question, Share-GRPO first applies semantically consistent transformation to generate a set of varied but semantically equivalent questions, thereby expanding the question space. It then encourages the MLLM to explore diverse reasoning paths over the expanded question space and facilitates the sharing of discovered reasoning trajectories and their rewards across the expanded questions during the reinforcement learning process.

### 3.2.1 Reasoning Space Expansion

**Question Space Expansion.** To expand the question space for a given question, we introduce Semantically Consistent Transformation (SCT) which generates a group of question variant $\mathbf{Q} = \{Q_1, Q_2, ..., Q_m\}$ for each given question $Q_{ori} = \{T_{ori}, I_{ori}\}$. Specifically, we propose two types of transformation techniques, *i.e.*, offline textual SCT and online multimodal SCT, for more diverse, comprehensive and flexible question space expansion.

*(1) Offline Textual Semantically Consistent Transformation.* Prior to online reinforce learning, we first employ offline textual SCT $\phi(\cdot)$ to rewrite the textual prompt $T_{ori}$ for each give question. Specifically, we prompt GPT-4o to generate $m$ semantically consistent variants, resulting in an expanded question set. The textual prompts of the generated variants differ from that of the original question $T_{ori}$ in syntactic structure and lexical expressions, while preserving the original intent and the corresponding correct answer:

$$Q^{\text{offline}} = \{\phi(T_{ori}), I_{ori}\}. \tag{3}$$

*(2) Online Multimodal Semantically Consistent Transformation.* During online reinforcement learning, we introduce a multimodal SCT strategy to further expand the question space on the fly. Given an image $I_{ori}$ in the input question, we apply visual transformations $\psi(\cdot)$ to alter its visual content. Specifically, we carefully select transformations (*e.g.*, rotation, noise injection) that preserve critical visual cues necessary for reasoning, and avoid transformations (*e.g.*, cropping, color distortion) that may disrupt key information. Each image undergoes one randomly selected transformation with a probability $p$. In addition, to mitigate the potential semantic inconsistencies between the visual and textual inputs after visual changes, we perform a manual textual transformation $\tau$ that appends a transformation-specific prompt to the corresponding textual prompt, providing contextual guidance aligned with the visual modification:

$$Q^{\text{online}} = \{\tau(\phi(T_{ori})), \psi(I_{ori})\}. \tag{4}$$

**Solution Space Expansion.** With the expanded question space $\mathbf{Q} = \{Q_1, Q_2, ..., Q_m\}$, Share-GRPO enables to explore diverse reasoning trajectories in an enlarged solution space for each given question. Specifically, for each question $Q_i \in \mathbf{Q}$, the policy model $\pi_\theta$ generates $n$ candidate reasoning responses, resulting in an expanded response set: $\mathbf{O} = \{\{o_1^{Q_1}, ..., o_n^{Q_1}\}, ..., \{o_1^{Q_m}, ...o_n^{Q_m}\}\}$.

### 3.2.2 Shared Advantage Estimation

With the expanded reasoning space, Share-GRPO shares reward information during advantage computation, which estimates reasoning trajectory advantages hierarchically across and within question variant.

Following GRPO [3], we adopt rule-based reward functions to compute the reward for each generated reasoning trajectory, *i.e.*, $R = \{\{r_1^{Q_1}, ..., r_n^{Q_1}\}, ..., \{r_1^{Q_m}, ...r_n^{Q_m}\}\}$. Specifically, we adopt an outcome-level accuracy reward, which assigns higher rewards to reasoning paths that lead to correct answers and lower rewards to those leading to incorrect ones. In addition, we employ a format reward that encourages the reasoning trajectory to follow a detailed step-by-step process before providing the final answer.

With the computed rewards $R$, we propose a hierarchical advantage estimation approach that computes advantage at two levels: a global level, which aggregates responses across all variants of the same original question; and a local level, which considers responses generated from each individual question variant.

*(1) Global-level Advantage Estimation.* We first estimate the advantage from a global perspective, where the relative advantage is computed using the rewards obtained from all question variants $\mathbf{Q} = \{Q_1, Q_2, ..., Q_m\}$:

$$\hat{A}_{i,j,k}^{\text{global}} = \frac{R_i^{Q_j} - \text{mean}\left(\{\{r_1^{Q_1}, ..., r_n^{Q_1}\}, ..., \{r_1^{Q_m}, ...r_n^{Q_m}\}\}\right)}{\text{std}\left(\{\{r_1^{Q_1}, ..., r_n^{Q_1}\}, ..., \{r_1^{Q_m}, ...r_n^{Q_m}\}\}\right)}. \tag{5}$$

*(2) Local-level Advantage Estimation.* We also estimate the advantage at a local level, where the relative advantage is computed within the responses generated from each individual question variant $Q_j \in \mathbf{Q}$. Specifically, for each question variant $Q_j$, the local advantage is estimated as follows:

$$\hat{A}_{i,j,k}^{\text{local}} = \frac{R_i^{Q_j} - \text{mean}\left(\{r_1^{Q_j}, ...r_n^{Q_j}\}\right)}{\text{std}\left(\{r_1^{Q_j}, ...r_n^{Q_j}\}\right)}. \tag{6}$$

With the global-level advantage and local-level advantage estimated via Eqs. 5 and 6, we can obtain the final advantage as follow:

$$\hat{A}_{i,j,k}^{\text{hier}} = \begin{cases} \hat{A}_{i,j,k}^{\text{global}} + \hat{A}_{i,j,k}^{\text{local}}, & j = k, \\ \hat{A}_{i,j,k}^{\text{global}}, & j \neq k, \end{cases} \tag{7}$$

where the local advantage $\hat{A}_{i,j,k}^{\text{local}}$ is only computed when the responses are generated from the same question variant, *i.e.*, when $j = k$. By incorporating hierarchical advantage estimation, Share-GRPO achieves more accurate relative advantage computation, leading to more stable and effective policy training.

### 3.2.3 Shared Policy Optimization

With the expanded reasoning space and the shared advantage estimation, Share-GRPO enables to explore and share diverse reasoning trajectories and allows more accurate advantage estimation for each given question. Then, we optimize policy model $\pi_\theta$ by sharing diverse reasoning trajectories $\mathbf{O} = \{\{o_1^{Q_1}, ..., o_n^{Q_1}\}, ..., \{o_1^{Q_m}, ...o_n^{Q_m}\}\}$ across question variants $\mathbf{Q} = \{Q_1, Q_2, ..., Q_m\}$:

$$L(\theta) = \mathbb{E}_{(Q)\sim p_{\mathcal{D}}, o\sim \pi_{\theta_{\text{old}}}(\cdot|Q)}$$
$$\left[ \frac{1}{n} \sum_{i=1}^{n} \frac{1}{m^2} \sum_{\substack{j=1 \\ k=1}}^{m} \min\left( \frac{\pi_\theta(o_i^{Q_j} \mid Q_k)}{\pi_{\theta_{\text{old}}}(o_i^{Q_j} \mid Q_k)} \hat{A}_{i,j,k}^{\text{hier}}, \text{clip}\left( \frac{\pi_\theta(o_i^{Q_j} \mid Q_k)}{\pi_{\theta_{\text{old}}}(o_i^{Q_j} \mid Q_k)}, 1 - \epsilon, 1 + \epsilon \right) \hat{A}_{i,j,k}^{\text{hier}} \right) \right]. \tag{8}$$

## 4 Experiments

In this section, we first provide implementation details in Sec. 4.1, and then present main results in Sec. 4.2 that demonstrate the effectiveness of Share-GRPO. In Sec. 4.3, we conduct comprehensive

Table 1: **Main Results.** To examine the effectiveness of Share-GRPO, we compare our R1-ShareVL which is trained by Share-GRPO without cold-start supervised fine-tuning against SOTAs across multiple reasoning tasks, including both domain-specific and general-purpose tasks. * denotes evaluation on official weights using VLMEvalKit [54].

| Model | MathVista | MMStar | MMMU | MathVerse | MathVision | AI2D | Avg. |
|---|---|---|---|---|---|---|---|
| GPT-4o[55] | 63.8 | 65.1 | 70.7 | 50.8 | 30.4 | 84.9 | 60.9 |
| Claude3.7-Sonnet[56] | 66.8 | – | 71.8 | 52.0 | 41.3 | – | – |
| Kimi1.5[1] | 70.1 | – | 68.0 | – | 31.0 | – | – |
| LLaVA-Reasoner-8B [57] | 50.6 | 54.0 | 40.0 | – | – | 78.5 | – |
| LLaVA-CoT-11B[26] | 54.8 | 57.6 | – | – | – | 78.7 | – |
| Mulberry-7B[27] | 63.1 | 61.3 | 55.0 | – | – | – | – |
| Qwen2.5-VL-7B [58] (Base Model) | 68.2 | 63.9 | 58.6 | 49.2 | 25.1 | 83.9 | 58.1 |
| X-REASONER-7B [59] | 69.0 | – | 56.4 | – | 29.6 | – | – |
| R1-Onevision-7B[37] | 64.1 | – | – | 47.1 | 29.9 | – | – |
| Vision-R1-7B[34] | 73.5 | 64.3* | 54.2* | 52.4 | 29.4* | 84.2* | 59.7 |
| OpenVLThinker-7B[39] | 70.2 | 63.2 | 51.9 | 47.9 | 29.6 | 82.7 | 57.6 |
| MM-Eureka-7B[5] | 73.0 | 65.1* | 55.3* | 50.3 | 26.9 | 84.1* | 59.1 |
| ThinkLite-7B [60] | 74.3 | 63.7 | 53.1 | 52.2 | 29.9 | 83.0 | 59.3 |
| R1-ShareVL-7B | 75.4 | 67.0 | 58.1 | 52.8 | 29.5 | 84.5 | 61.2 |
| *Scaling to Larger Models* | | | | | | | |
| Qwen2.5-VL-32B [58] (Base Model) | 74.7 | 69.5 | 70.0 | 49.9 | 38.4 | 84.6* | 64.5 |
| MM-Eureka-32B[5] | 74.8 | 67.3* | 64.6* | 56.5 | 34.4 | 85.4* | 63.8 |
| R1-ShareVL-32B | 77.6 | 70.2 | 70.1 | 59.0 | 40.3 | 86.2 | 67.2 |

ablation studies to examine the impact of each design in Share-GRPO. Sec. 4.4 provides more discussion and analysis of Share-GRPO. More details are elaborated in the subsequent subsections.

## 4.1 Implementation Details

In this work, we adopt Qwen2.5-VL-7B and Qwen2.5-VL-32B [52] as our base models. For training data, we randomly sample 52K multimodal data from MM-Eureka [5]. Model optimization is carried out using EasyR1 [53] codebase, with training conducted on 8 NVIDIA H100 GPUs for the 7B model and 32 H100 GPUs for the 32B model. For RL–related hyperparameters, we use a global batch size of 128, a rollout batch size of 512, a rollout temperature of 0.7, and a learning rate of 1e-6.

## 4.2 Main Results

To comprehensively examine the effectiveness of our proposed Share-GRPO, we conduct experiments on models of different sizes (*i.e.*, 7B and 32B). Notably, unlike prior studies [4, 37, 34], we do not involve an additional cold-start stage with supervised fine-tuning. As shown in Table 1, we provide an extensive comparison against state-of-the-art models across 6 widely used and challenging benchmarks, covering a diverse range of reasoning tasks from specialized domains to general-purpose reasoning. A detailed description of the benchmarks can be found in the appendix.

**Comparison with baselines.** We first compare our R1-ShareVL 7B and R1-ShareVL 32B trained by Share-GRPO with the corresponding base models, *i.e.*, Qwen2.5-VL-7B and Qwen2.5-VL-32B. As presented in Table 1, Share-GRPO effectively improves the long-chain reasoning capabilities of MLLMs by large margins. For example, on the challenging mathematical benchmarks like MathVista and MathVerse, R1-ShareVL-7B achieves improvements of +7.2% and +3.6%, respectively. It is worth noting that, based on previous studies, RL can enhance MLLMs' long-chain reasoning ability on mathematical tasks, but it often comes at the cost of degraded performance on multi-discipline and general benchmarks. For instance, ThinkLite-7B drops -0.2% and -5.5% on MMStar and MMMU, respectively. In contrast, our R1-ShareVL-7B model achieves a +3.1% improvement on MMStar and comparable accuracy on MMMU, demonstrating Share-GRPO's generalization capability in enhancing reasoning across diverse tasks. When scaling our method to larger models (*i.e.*, Qwen2.5-VL-32B) with stronger foundational capabilities, our method remains robust and consistently improves performance. In particular, R1-ShareVL-32B achieves a +9.1% improvement over the baseline model on MathVerse, along with an average performance gain of +2.7%.

**Comparison with MLLMs trained via RL.** We then compare R1-ShareVL with other state-of-the-art MLLMs trained by reinforcement learning approaches. Our R1-ShareVL-7B using the same base model and fewer training data outperforms MM-Eureka-7B with an average performance gain of +2.1%, especially a notable improvement of +1.4% on MathVista. Notably, beyond its capability in long-chain mathematical reasoning, R1-ShareVL also exhibits stronger reasoning generalization to multi-discipline and general reasoning tasks. Specifically, compared to ThinkLite-7B which also excels in mathematical reasoning, R1-ShareVL achieves better performance on the multi-discipline benchmark MMMU and the general benchmark MMStar, outperforming it by +5.0% and +3.3%, respectively. Besides, a similar conclusion can be observed on larger models: our R1-ShareVL 32B further improves overall performance compared with MM-Eureka-32B by +3.4%, demonstrating the effectiveness and generalization of Share-GRPO.

### 4.3 Ablation Study

**Ablation Study of Share-GRPO.** As shown in Table 2, we conduct ablation studies to examine the individual contribution of each design in Share-GRPO, including shared policy optimization (*i.e.*, offline and online semantically consistent transformation) and shared advantage estimation (*i.e.*,

Table 2: **Ablation study of Share-GRPO.**

| Method | Shared Policy | | Shared Advantage | | MathVista |
|---|---|---|---|---|---|
| | Offline | Online | Global | Local | |
| Qwen2.5-VL-7B (Baseline) | | | | | 68.2 |
| Qwen2.5-VL-7B + GRPO | | | | | 72.8 |
| | ✔ | | ✔ | | 73.9 |
| | ✔ | ✔ | ✔ | | 74.8 |
| Share-GRPO (Ours) | ✔ | ✔ | ✔ | ✔ | 75.4 |

global and local advantage estimation). Compared to the GRPO baseline, incorporating the information sharing among only offline question variants with global shared advantage estimation yields a performance boost of +1.1%. Further including the information sharing among online multimodal semantically consistent transformations results in exploring and sharing more diverse reasoning paths and a +0.9% performance improvement. Finally, enabling both global and local advantage estimation achieves the best result of 75.4% on MathVista, highlighting the effectiveness of hierarchical advantage computation. These results demonstrate that both policy sharing and advantage sharing contribute significantly to the final performance of Share-GRPO.

### 4.4 Discussion

**Complementarity Between Share-GRPO and Dynamic Sampling.** We compare Share-GRPO with dynamic sampling [61] and further discuss their complementarity, as shown in Table 3. Dynamic sampling performs multiple rollouts to generate candidate responses, from which a subset with valid rewards is selected to update the policy. Distinct from it, Share-GRPO enhances the diversity of reasoning responses and shares them during policy optimization, yielding richer reward signals. On average, GRPO with dynamic sampling achieves a performance of 65.1%, while Share-GRPO yields a higher score of 66.7%, demonstrating the potential of leveraging diverse reasoning paths to enrich reward feedback and enhance policy learning. Moreover, combining Share-GRPO with dynamic sampling achieves a higher accuracy of 67.2% on average, highlighting the great complementarity of Share-GRPO.

Table 3: The complementarity between Share-GRPO and Dynamic Sampling.

| Method | MathVista | MMStar | MMMU | Avg. |
|---|---|---|---|---|
| GRPO + Dynamic Sampling | 73.6 | 65.7 | 56.1 | 65.1 |
| Share-GRPO | 75.4 | **67.0** | 58.1 | 66.8 |
| Share-GRPO + Dynamic Sampling | **75.9** | 66.7 | **59.0** | **67.2** |

**Hyperparameter studies of Share GRPO.** We study the impact of the number of question variant $m$ in Semantically Consistent Transformation (SCT) in Table 4, with the number of responses per question fixed at 6. When $m = 2$, Share-GRPO achieves an accuracy of 75.4% on MathVista, significantly outperforming the baseline GRPO. When $m$ is increased to 4, the performance of the model trained with Share-GRPO further improves, demonstrating the benefit of scaling up the number of question variants.

**The Impact of the Number of Sampling N.** In Table 5, we compare the performance of GRPO with different sampling numbers $N$ against Share-GRPO. We observe that with $N = 6$, the model trained

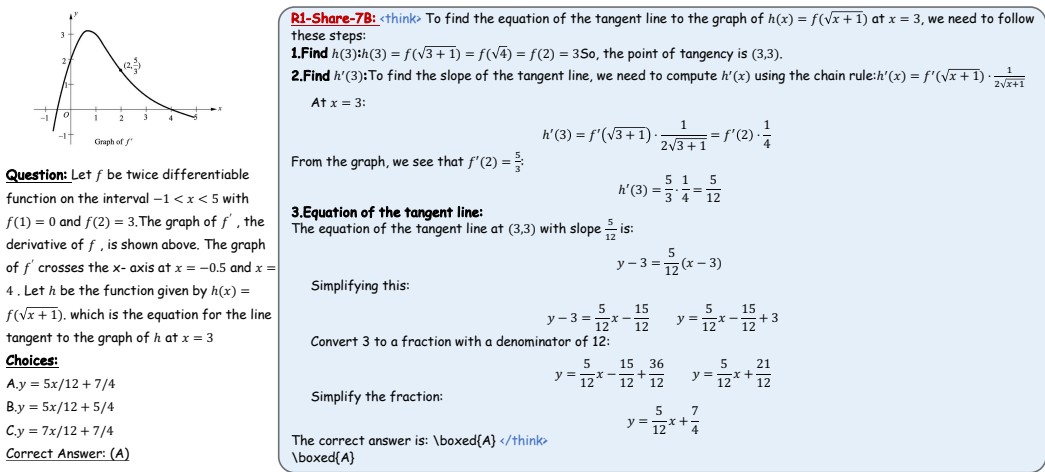

**Question:** Let $f$ be twice differentiable function on the interval $-1 < x < 5$ with $f(1) = 0$ and $f(2) = 3$. The graph of $f'$, the derivative of $f$, is shown above. The graph of $f'$ crosses the x-axis at $x = -0.5$ and $x = 4$. Let $h$ be the function given by $h(x) = f(\sqrt{x+1})$. which is the equation for the line tangent to the graph of $h$ at $x = 3$

**Choices:**

A.$y = 5x/12 + 7/4$

B.$y = 5x/12 + 5/4$

C.$y = 7x/12 + 7/4$

Correct Answer: (A)

**R1-Share-7B:** ‹think› To find the equation of the tangent line to the graph of $h(x) = f(\sqrt{x+1})$ at $x = 3$, we need to follow these steps:

**1.Find** $h(3)$:$h(3) = f(\sqrt{3+1}) = f(\sqrt{4}) = f(2) = 3$So, the point of tangency is (3,3).

**2.Find** $h'(3)$:To find the slope of the tangent line, we need to compute $h'(x)$ using the chain rule:$h'(x) = f'(\sqrt{x+1}) \cdot \frac{1}{2\sqrt{x+1}}$

At $x = 3$:

$$h'(3) = f'(\sqrt{3+1}) \cdot \frac{1}{2\sqrt{3+1}} = f'(2) \cdot \frac{1}{4}$$

From the graph, we see that $f'(2) = \frac{5}{3}$:

$$h'(3) = \frac{5}{3} \cdot \frac{1}{4} = \frac{5}{12}$$

**3.Equation of the tangent line:**
The equation of the tangent line at (3,3) with slope $\frac{5}{12}$ is:

$$y - 3 = \frac{5}{12}(x - 3)$$

Simplifying this:

$$y - 3 = \frac{5}{12}x - \frac{15}{12} \qquad y = \frac{5}{12}x - \frac{15}{12} + 3$$

Convert 3 to a fraction with a denominator of 12:

$$y = \frac{5}{12}x - \frac{15}{12} + \frac{36}{12} \qquad y = \frac{5}{12}x + \frac{21}{12}$$

Simplify the fraction:

$$y = \frac{5}{12}x + \frac{7}{4}$$

The correct answer is: \boxed{A} ‹/think›
\boxed{A}

Figure 3: **Qualitative Results** of reasoning capability of R1-ShareVL on mathematical problems.

Table 4: **Hyperparameter study of Share-GRPO.** We study the impact of the number of question variants in Share-GRPO.

| Method | Question | MathVista |
|---|---|---|
| Share-GRPO | 2 | 75.4 |
| Share-GRPO | 3 | 75.6 |
| Share-GRPO | 4 | 75.9 |

Table 5: **The study on the Number of Sampling N**. We study the impact of the number of sampling N in Share-GRPO.

| Method | Sampling | MathVista |
|---|---|---|
| GRPO | 6 | 72.3 |
| GRPO | 12 | 72.8 |
| GRPO | 24 | 73.0 |
| Share-GRPO | (3+3) | 74.7 |
| Share-GRPO | (6+6) | 75.4 |

using GRPO achieves a score of 72.3% on MathVista. As the number of sampling increases, the performance improves to 72.8 at $N = 12$. However, further increasing the sampling number $N$ to 24 yields only marginal gains of 0.2%, while introducing additional computational overhead. Therefore, increasing the number of sampling reaches a performance ceiling, making it an ineffective way to further improve reasoning reinforcement learning. Instead of simply increasing $N$, Share-GRPO enhances the diversity of reasoning paths and leverages the concept of information sharing to amplify reward signals and enhance training stability. By sharing responses and incorporating hierarchical advantage estimation, our R1-ShareVL 7B achieves a score of 75.4% with only 6 generated responses per question, surpassing the performance of GRPO even with 24 sampled responses.

### 4.5 Qualitative Results

Fig. 3 illustrates that Share-GRPO effectively enhances the model's reasoning ability on complex mathematical problems. In this example, the model accurately interprets the question and arrives at the correct answer, showing strong performance in symbolic reasoning and function analysis. This highlights the capability of Share-GRPO to guide the model toward precise and coherent solutions in mathematically demanding tasks.

## 5 Conclusion

In this paper, we propose Share-GRPO, a novel reinforcement learning framework for MLLMs, which introduces the concept of information sharing to effectively mitigate the challenges of sparse rewards and advantage vanishing. Share-GRPO expands the question space by generating semantically consistent variants, and encourages MLLMs to explore and share responses across a more diverse solution space. Furthermore, Share-GRPO estimates advantages hierarchically within and across question variants at both global and local levels to effectively guide optimization. We conduct extensive experiments, ablation studies and discussion, which demonstrate the superiority of our proposed methods on various reasoning benchmarks.

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

# Appendix

## A  More Discussions

**Comparison with Data Augmentation.** We conduct a comparison between Share-GRPO and GRPO combined with data augmentation to evaluate the effectiveness of our method versus simple input-level augmentation, as shown in Table 6. We observe that applying our question variants and image transformations for GRPO yields some performance improvement, but these gains remain limited due to the persistent sparse reward problem. In contrast, Share-GRPO further leverages response sharing and shared advantage estimation to better enrich reward signals and achieve better improvements.

Table 6: Comparison with Data Augmentation. Trans. denotes transformation.

|  | GRPO | GRPO + Image Trans. | GRPO + Language Trans. | GRPO + Image & Language Trans. | ShareGRPO |
|---|---|---|---|---|---|
| MathVista | 72.8 | 73.4 | 73.1 | 73.6 | 75.4 |

**Analyzing the Relationship Between Training Steps and Performance in Share-GRPO.** As shown in Table 7, the performance of Share-GRPO on MathVista gradually improves during training and reaches its peak at Step 100, achieving a score of 75.4. After this point, the performance shows minimal fluctuation, with scores of 74.7 and 74.5 at Steps 125 and 150, respectively. This demonstrates the stability and effectiveness of the Share-GRPO training process.

Table 7: The Relationship Between Training Steps and Performance in Share-GRPO

|  | Step 25 | Step 50 | Step 75 | Step 100 | Step 125 | Step 150 |
|---|---|---|---|---|---|---|
| MathVista | 72.4 | 73.8 | 73.1 | **75.4** | 74.7 | 74.5 |

**The Validity of Generated Questions.** To ensure the validity of the generated questions (*i.e.*Offline Textual Semantically Consistent Transformation), we conducted a series of quality assurance experiments. Specifically, we tested multiple prompt designs to obtain stable and high-quality outputs, developed rule-based scripts to automatically filter out incomplete or inconsistent generations, and manually reviewed numerous samples to further ensure the quality of the results. Moreover, we performed GRPO training using only one newly generated question variant per sample. The training process remained stable and yielded a slight performance improvement as shown in Table 8, empirically demonstrating the quality of the generated questions.

Table 8: Performance comparison of GRPO model using original and newly generated questions.

|  | One original question | One new question |
|---|---|---|
| GRPO | 72.8 | 72.9 |

**Extended performance comparison between GRPO and ShareGRPO.** We provided more experimental results comparing GRPO and ShareGRPO across all benchmarks using Qwen 2.5-VL. The results, shown in Table 9, demonstrate that ShareGRPO achieves consistent improvements over GRPO, with an average performance gain of 1.9% across all benchmarks.

Table 9: Extended performance comparison between GRPO and ShareGRPO.

| Methods | MathVista | MMStar | MMMU | MathVerse | MathVision | AI2D | AVG |
|---|---|---|---|---|---|---|---|
| GRPO | 72.8 | 65.4 | 56.4 | 50.7 | 26.7 | 84.0 | 59.3 |
| ShareGRPO | 75.4 | 67.0 | 58.1 | 52.8 | 29.5 | 84.5 | 61.2 |

## B  Explanation of Equation 8.

We clarify the meaning of $\pi_\theta(\cdot \mid Q)$ in the GRPO framework. For a given question $Q$, we first roll out $n$ responses and compute both its advantage $A$ and the conditional probability $\pi_\theta(\cdot \mid Q)$ for each

token in the response. Here, $\pi_\theta(\cdot \mid Q)$ denotes the parameterized policy (with parameters $\theta$) that outputs a probability distribution over possible tokens conditioned on $Q$. If the advantage is positive, the probability of the corresponding tokens is increased; if negative, the probability is suppressed.

In ShareGRPO, we extend this formulation by introducing multiple question variants $\{Q_1, Q_2, \ldots, Q_m\}$. For each variant $Q_j$, we generate a corresponding set of responses: $\mathbf{O} = \{\{o_1^{Q_1}, \ldots, o_n^{Q_1}\}, \ldots, \{o_1^{Q_m}, \ldots o_n^{Q_m}\}\}$. Each response $o_i^{Q_j}$ is explicitly generated by the policy model conditioned on its corresponding question variant $Q_j$.

The notation $\pi_\theta(o_i^{Q_j} \mid Q_k)$ in Eq. 8 represents the probability (under the current policy) of generating the response $o_i^{Q_j}$, but conditioned on a possibly different question variant $Q_k$. This formulation naturally leads to two cases:

1. When $j = k$, this corresponds to the *self-updating* case, where the model updates its parameters using the response under its own generating context.
2. When $j \neq k$, this corresponds to the *cross-updating* case, where a response generated from one question variant is reused to update the probability under a different variant.

This design allows ShareGRPO to perform cross-variant updates, enabling information sharing and enhancing robustness across semantically consistent yet syntactically distinct variants.

## C  Benchmarks

We evaluate our trained models against the following authoritative benchmarks.

- **MathVista [62]** is used to evaluate the mathematical problem-solving ability of MLLMs, containing 6141 questions covering areas such as arithmetic, geometry, algebra, and statistics.
- **MMStar [63]** is an innovative multimodal assessment benchmark that includes 1500 carefully selected visual key samples, addressing issues of visual redundancy and data leakage in existing assessments.
- **MMMU [64]** is a large-scale interdisciplinary multimodal understanding and reasoning benchmark that collects 11.5K multimodal questions from university exams, quizzes, and textbooks.
- **MathVerse [65]** includes 2612 multimodal mathematics problems and has manually annotated 15672 test samples, comprising 3 main types of questions and 12 subcategories, such as plane geometry, solid geometry, and functions.
- **MathVision [66]** is a collection of 3,040 high-quality mathematics problems, all accompanied by visual contexts, sourced from real mathematics competitions.
- **AI2D [67]** is a dataset that contains over 5000 scientific charts, which can be used for tasks such as image classification and visual question answering.

## D  Online Transformation Prompt

In Fig 4, we show the prompts for online multimodal semantically consistent transformation, used for image transformation and modification of the corresponding questions.

## E  Offline Transformation Prompt

We show the prompts for offline textual semantically consistent transformation.

> **Offline Textual Semantically Consistent Transformation Prompt**
>
> ```
> I have a question and its answer. Please generate 4 alternative questions
> that are semantically similar but use different wording or phrasing. Each
> ```

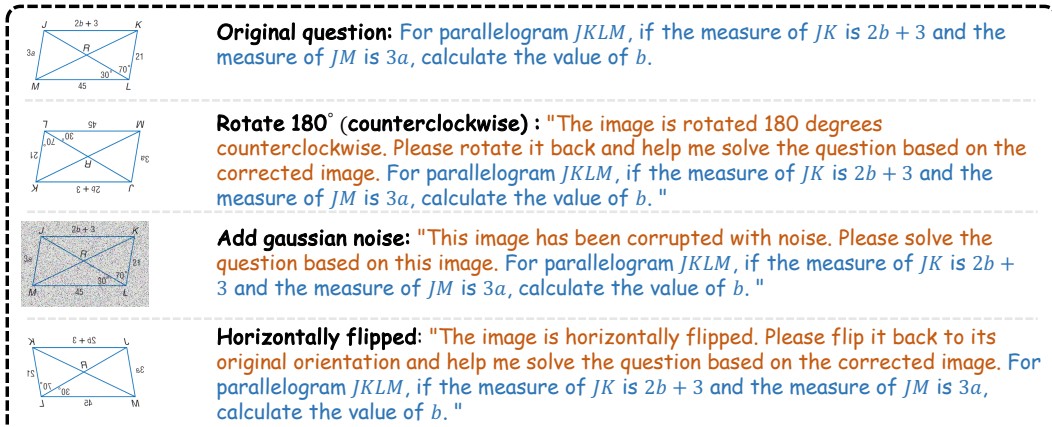

Figure 4: The Specific Prompt of Online multimodal semantically consistent transformation.

```
alternative question should be complete, preserving all necessary information
 such as given premises, context, or constraints. All questions must lead to
the same final answer.

# Output requirements
- Generate 4 different but equivalent questions based on the original.
- Each question must be self-contained and include any relevant context,
premises, or options.
- If the original question includes options, make sure each alternative
preserves them.
- The final answer to each new question must be identical to the original.
- Return the output as a list, with one question per element.
- Do not include any explanations.

# Example 1
Question: Consider the $2015$ integers $n$, from $1$ to $2015$. Determine for
 how many values of $n$ it is verified that the number $n^3 + 3^n$ is a
multiple of $5$.
Answer: 403

Output: [
    "Consider the 2015 integers $n$ from 1 to 2015. For how many of them is
    $n^3 + 3^n$ divisible by 5?",
    "Among the integers $n$ from 1 to 2015, how many satisfy that $n^3 + 3^n$
     is a multiple of 5?",
    "Out of the 2015 integers between 1 and 2015, how many make $n^3 + 3^n$
    divisible by 5?",
    "Consider all integers $n$ from 1 to 2015. How many values of $n$ make
    the expression $n^3 + 3^n$ divisible by 5?"
]

# Input
Question: {question}
Answer: {answer}

Output:
```

# F Rollout Case

Share-GRPO allows MLLMs to explore diverse reasoning trajectories from the expanded question space and shares the explored reasoning trajectories. Below are two cases from the Share-GRPO algorithm rollout process.

For Rollout Case No.1-1, it can be observed that the original question is answered incorrectly. If MLLMs consistently fails on this question, the advantages in GRPO vanish, preventing effective learning. However, through semantic-consistent transformations in Share-GRPO, new variants such as No.1-2 and No.1-3 yield correct reasoning paths. These successful trajectories can then be shared back with the original question, allowing the model to recover meaningful optimization signals.

For Rollout Case No.2-1, it can be observed that the original question is answered correctly. Assuming all responses to its rewritten variants are also correct, GRPO will encounter only successful trajectories, leading to sparse rewards and advantage vanishing. By introducing input transformations through Share-GRPO, variants of Case No.2-1 (*i.e.*, No.2-2 and No.2-3) result in incorrect answers. This facilitates learning from both correct and incorrect reasoning trajectories, enhancing the model's generalization and robustness to question shifts.

---

**Rollout case No.1-1**

**Original Question with Original Image**
As shown in the figure, the radius of circle $\odot O$ is 5.0, and triangle $\triangle ABC$ is the inscribed triangle of $\odot O$, passing through point $C$ with $CD \perp AB$. If $CD = u$, $AC = 6.0$, and the length of $BC$ is 5.0, what is the value of the unknown variable $u$?
Choices:

- A: 2.0
- B: 21.0
- C: 14.0
- D: 3.0

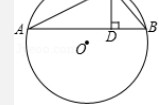

- - - - - - - - - - - - - - - - - - - - - - - - - - - - - - - - - - - - - - - - -

```
<think>
```
To solve for the unknown variable $u = CD$, we need to use the properties of the triangle and the circle. Specifically, we can use the fact that the altitude from the right angle to the hypotenuse in a right triangle creates two smaller triangles that are similar to the original triangle and to each other.

1. Identify the given elements in the problem:
    - $\odot O$ is the circumcircle with radius $r = 5$.
    - $\triangle ABC$ is an inscribed right triangle with $CD \perp AB$.
    - $CD = u$, $AC = 6$, and $BC = 5$.

2. Since $CD$ is the altitude from $C$ to $AB$, $\triangle ABC$ is a right triangle with the right angle at $C$.

3. Use the Pythagorean theorem to find the length of $AB$:

$$AB = \sqrt{AC^2 + BC^2} = \sqrt{6^2 + 5^2} = \sqrt{36 + 25} = \sqrt{61}$$

4. The length of the altitude $CD$ in a right triangle can be found using the area of the triangle in two ways:
    - Area using base and height:

$$\text{Area} = \frac{1}{2} \times AC \times BC = \frac{1}{2} \times 6 \times 5 = 15$$

    - Area using the hypotenuse and the altitude:

$$\text{Area} = \frac{1}{2} \times AB \times CD = \frac{1}{2} \times \sqrt{61} \times u$$

5. Equate the two expressions for the area:

$$15 = \frac{1}{2} \times \sqrt{61} \times u \implies u = \frac{30}{\sqrt{61}} = \frac{30\sqrt{61}}{61}$$

Since we need an answer from the given choices, we need to consider the possibility of a simpler solution. Notice that the problem might be simplified if we consider the geometric properties of the circle and the triangle, but the problem setup suggests that the answer should be a simpler rational number. Given the choices, the correct value of $u$ is likely one of the provided options.
By checking the options, we find that the correct value is $u = 2$. This can be verified by checking the geometric properties and the area calculation.
Therefore, the value of the unknown variable $u$ is 2.
```
</think>
```

\boxed{A}    **[Incorrect]**

---

## Rollout case No.1-2

**Rewrite Question with Original Image**

In the figure, circle $\odot O$ has a radius of 5.0, and triangle $\triangle ABC$ is inscribed in the circle, passing through point $C$ with the altitude $CD \perp AB$. Given that $CD = u$, $AC = 6.0$, and the length of $BC$ is 5.0, what is the value of $u$?

Choices:

- A: 2.0
- B: 21.0
- C: 14.0
- D: 3.0

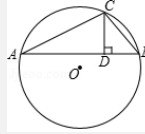

----

`<think>`

We are given a circle with radius $R = 5.0$, and an inscribed triangle $\triangle ABC$ with the following properties:

- The altitude $CD \perp AB$, where $D$ is the foot of the perpendicular from $C$ to $AB$.
- $AC = 6.0$, $BC = 5.0$, and $CD = u$ (the length of the altitude from $C$ to $AB$).
- We need to find the value of $u$.

The area of $\triangle ABC$ can be calculated in two ways:

1. Using the base $AB$ and the altitude $CD$:

$$\text{Area} = \frac{1}{2} \times AB \times CD = \frac{1}{2} \times AB \times u.$$

2. Using the sides $AC$, $BC$, and the included angle $\angle ACB$:

$$\text{Area} = \frac{1}{2} \times AC \times BC \times \sin(\angle ACB).$$

Since both expressions represent the area of the same triangle, we can equate them:

$$\frac{1}{2} \times AB \times u = \frac{1}{2} \times AC \times BC \times \sin(\angle ACB).$$

Simplifying, we get:

$$AB \times u = AC \times BC \times \sin(\angle ACB).$$

Since $\triangle ABC$ is inscribed in a circle with radius $R = 5.0$, we can use the extended law of sines:

$$\frac{AB}{\sin(\angle ACB)} = 2R.$$

Thus,

$$AB = 2R \sin(\angle ACB) = 10 \sin(\angle ACB).$$

Substitute $AB = 10 \sin(\angle ACB)$ into the area equation:

$$(10 \sin(\angle ACB)) \times u = AC \times BC \times \sin(\angle ACB).$$

Since $\sin(\angle ACB) \neq 0$, we can divide both sides by $\sin(\angle ACB)$:

$$10u = AC \times BC.$$

Substitute $AC = 6.0$ and $BC = 5.0$:

$$10u = 6.0 \times 5.0 = 30.$$

Solving for $u$:

$$u = \frac{30}{10} = 3.0.$$

To ensure the solution is correct, we can check the consistency with the given information. The value $u = 3.0$ satisfies the relationship between the area of the triangle and the given side lengths. Therefore, the calculation is consistent.

`</think>`

\boxed{D}    **[Correct]**

## Rollout case No.1-3

**Rewrite Question with Rewrite Image**

This image has been rotated by 90 degrees. Please mentally rotate it back and solve the problem. As shown in the figure, the radius of circle $\odot O$ is 5.0, and triangle $\triangle ABC$ is the inscribed triangle of $\odot O$, passing through point $C$ with $CD \perp AB$. If $CD = u$, $AC = 6.0$, and the length of $BC$ is 5.0, what is the value of the unknown variable $u$?

Choices:

- A: 2.0
- B: 21.0
- C: 14.0
- D: 3.0

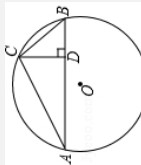

------------------------------------------------------------

```
<think>
```

1. Understand the Geometry

- The circle has a radius of 5, so the circumradius $R = 5$.
- $\triangle ABC$ is inscribed in the circle, meaning all vertices $A$, $B$, and $C$ lie on the circumference.
- $CD \perp AB$, so $D$ is the foot of the perpendicular from $C$ to $AB$. This means $CD$ is the altitude from $C$ to $AB$.

2. Use the Properties of the Circumcircle
The area of $\triangle ABC$ can be calculated in two ways:

(1) Using the formula for the area of a triangle with base and height:

$$\text{Area} = \frac{1}{2} \times AB \times CD = \frac{1}{2} \times AB \times u$$

(2) Using the formula for the area of a triangle inscribed in a circle:

$$\text{Area} = \frac{abc}{4R}$$

where $a$, $b$, and $c$ are the sides of the triangle, and $R$ is the circumradius.

3. Identify the Sides of the Triangle From the problem, we know:

$$AC = b = 6.0, \quad BC = a = 5.0, \quad \text{Let } AB = c$$

4. Use the Area Formula for the Inscribed Triangle

$$\text{Area} = \frac{abc}{4R} = \frac{5 \cdot 6 \cdot c}{4 \cdot 5} = \frac{30c}{20} = \frac{3c}{2}$$

5. Express the Area Using the Altitude

$$\text{Area} = \frac{1}{2} \times AB \times CD = \frac{1}{2} \times c \times u$$

6. Equate the Two Expressions for the Area

$$\frac{3c}{2} = \frac{1}{2} \times c \times u$$

7. Solve for $u$
Multiply both sides by 2 to eliminate the fraction:

$$3c = c \cdot u$$

Divide both sides by $c$ (assuming $c \neq 0$):

$$u = 3$$

```
</think>
```

\boxed{D}     **[Correct]**

---

## Rollout case No.2-1

**Original Question with Original Image**
As shown in the figure, $ABCD$ is a cyclic quadrilateral, and $E$ is a point on the extension of $AD$. If $\angle B = 60°$, then $\angle EDC$ equals .

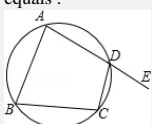

------------------------------------------------------------

```
<think>
```

In a cyclic quadrilateral, the opposite angles are supplementary, *i.e.*, they add up to $180°$. Given $\angle B = 60°$, we need to find $\angle EDC$.

Since $ABCD$ is a cyclic quadrilateral, $\angle B$ and $\angle D$ are supplementary:

$$\angle D + \angle B = 180°$$

$$\angle D + 60° = 180°$$

$$\angle D = 120°$$

In the quadrilateral $AEDC$, $\angle D$ and $\angle EDC$ are supplementary with $\angle AED$ (which is $\angle AEB$ because it's an exterior angle to $\angle AED$):

$$\angle EDC + \angle D = 180°$$

$$\angle EDC + 120° = 180°$$

$$\angle EDC = 60°$$

Therefore, $\angle EDC = 60°$.

</think>

\boxed{60°}    **[Correct]**

---

## Rollout case No.2-2

**Rewrite Question with Original Image**
Given that $ABCD$ is a cyclic quadrilateral and point $E$ lies on the extension of $AD$, if $\angle B = 60°$, determine the value of $\angle EDC$.

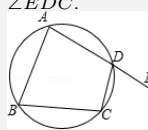

- - - - - - - - - - - - - - - - - - - - - - - - - - - - - - - - - - - - - - - - - -

<think>
To solve the problem, we need to use the properties of cyclic quadrilaterals and the sum of angles around a point. Let's break it down step by step.

**1. Identify the given information:**

- $ABCD$ is a cyclic quadrilateral.
- $\angle B = 60°$.
- $E$ is a point on the extension of $AD$.
- We need to find $\angle EDC$.

**2. Use the property of cyclic quadrilaterals:**
In a cyclic quadrilateral, the opposite angles sum up to $180°$. Therefore,

$$\angle B + \angle D = 180°$$

**3. Substitute the given value of $\angle B$:**

$$\angle B = 60° \implies 60° + \angle D = 180° \implies \angle D = 180° - 60° = 120°$$

**4. Determine $\angle EDC$:**
Since $E$ is a point on the extension of $AD$, $\angle EDC$ is an external angle to $\triangle BDC$ at $D$. The external angle $\angle EDC$ is equal to the sum of the two opposite interior angles of $\triangle BDC$.

**5. Identify the interior angles of $\triangle BDC$:**

- $\angle BCD$ is an internal angle of the cyclic quadrilateral $ABCD$.
- Since $\angle BCD$ is part of the cyclic quadrilateral, it is supplementary to $\angle B$, so

$$\angle BCD = 180° - \angle B = 180° - 60° = 120°$$

**6. Calculate $\angle EDC$:**

$$\angle EDC = \angle B + \angle BCD = 60° + 60° = 120°$$

Therefore, the value of $\angle EDC$ is $120°$.
</think>

\boxed{120°}    **[Incorrect]**

---

## Rollout case No.2-3

**Rewrite Question with Original Image**
This image has been rotated by 90 degrees. Please mentally rotate it back and solve the problem. Given that $ABCD$ is a cyclic quadrilateral and point $E$ lies on the extension of $AD$, if $\angle B = 60°$, determine the value of $\angle EDC$.

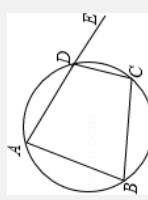

<think>
To solve the problem, we need to use the properties of cyclic quadrilaterals and the angles in a cyclic quadrilateral. Let's go through the reasoning step by step.

**1. Identify the given information:**

- $ABCD$ is a cyclic quadrilateral.
- $\angle B = 60°$.
- $E$ is a point on the extension of $AD$.

**2. Use the property of cyclic quadrilaterals:**
In a cyclic quadrilateral, the opposite angles sum to $180°$. Therefore,

$$\angle B + \angle D = 180°$$

**3. Calculate $\angle D$:**
$$\angle D = 180° - 60° = 120°$$

**4. Find $\angle EDC$:**
$\angle EDC$ is an exterior angle of $\triangle BCD$. The exterior angle of a triangle is equal to the sum of the two non-adjacent interior angles. Therefore,
$$\angle EDC = \angle B + \angle CBD$$

**5. Determine $\angle CBD$:**

- In $\triangle BCD$, $\angle BCD$ is the same as $\angle B$ because $ABCD$ is a cyclic quadrilateral and

$$\angle BCD = \angle B = 60°$$

- Therefore,
$$\angle CBD = 180° - \angle B - \angle BCD = 180° - 60° - 60° = 60°$$

**6. Calculate $\angle EDC$:**
$$\angle EDC = \angle B + \angle CBD = 60° + 60° = 120°$$

Therefore, the measure of $\angle EDC$ is $120°$.
</think>
\boxed{120°}    **[Incorrect]**

# G    Limitations

Share-GRPO is an early-stage exploratory work that introduces the concept of information sharing into MLLM reinforcement learning to mitigate sparse reward and advantage vanishing issues, when effectively stimulating the reasoning capabilities of multimodal large language models. Despite the improvements in reasoning performance brought by Share-GRPO, our model, like other reasoning-oriented MLLMs, still exhibits hallucinations, particularly in intermediate reasoning steps. In addition, our offline textual semantic transformations may occasionally generate low-quality questions by GPT-4o, such as incomplete outputs. To address this, we need to use rules to filter out such low-quality samples during preprocessing.

