# OpenReview forum: "R1-ShareVL: Incentivizing Reasoning Capabilities of Multimodal Large Language Models via Share-GRPO"
_NeurIPS.cc/2025/Conference — NeurIPS 2025 poster_

### Official Review · Reviewer_MP3K · 2025-06-05

**Clarity:** 4
**Significance:** 4
**Originality:** 3
**Rating:** 4
**Confidence:** 4

**Summary:**

This article addresses the problems of reward sparsity and advantage vanishing encountered when applying GRPO to the MLLM domain. According to the article, reward sparsity refers to the difficulty MLLMs have in generating correct answers, which makes it hard for the model to obtain positive training data with high rewards. Advantage vanishing means that the model often produces similar answers to the same question; since these similar answers receive comparable rewards, their relative advantage becomes very small. The article proposes expanding the questions and augmenting the images to enable the model to generate a more diverse set of answers, thereby achieving a better distribution of rewards and advantages, which facilitates model training.

**Questions:**

Please refer to the weakness section. This is a good paper, and I recommend acceptance. If the authors can address my concerns mentioned in the weakness section, I would be happy to raise my score.

**Ethical Concerns:**

["NO or VERY MINOR ethics concerns only"]

**Final Justification:**

This is a well-written article with a clear problem statement and a clear methodology. After our discussion, I realized that the author's experiments mainly focus on the mathematical domain. In fact, there should be more applications in other domains as well. I suggest conducting further research in more areas to provide a more solid conclusion.

**Limitations:**

yes

**Quality:**

4

**Strengths And Weaknesses:**

The advantages of this article are obvious.

[+] The article clearly identifies the drawbacks of GRPO, and these issues are indeed important to address.

[+] The proposed method is very effective and shows significant improvement compared to other approaches applying GRPO.

The shortcomings of the article are minor; I only have a few small questions:

[-] Why is the improvement not obvious on certain benchmarks, e.g., MMMU and AI2D ?

[-] The drawbacks of GRPO identified in the article do not seem to be limited to the multimodal setting. Could the shared-GRPO approach also be applied successfully in the NLP domain?

[-] The approach of augmenting images to enable the model to produce diverse answers is quite similar to the method in the paper [MM2024, oral] "Self-Supervised Visual Preference Alignment." Methods like SeVa, which use DPO for self-training, have shown significant performance gains. Would it be possible to consider comparing GRPO with DPO?

---

> ### Author Rebuttal · Authors · 2025-07-31
>
> Dear reviewer MP3K,
>
> Thank you for your professional and valuable feedback.
> We appreciate your recognition that our paper clearly identifies the drawbacks of GRPO and addresses these important issues. We are also encouraged by your positive assessment of our proposed method’s effectiveness and its significant improvements over other GRPO-based approaches.
> Below, we address your concerns and questions individually:
>
> ---
>
> **W1: Analysis of model improvements on certain benchmark, e.g., MMMU and AI2D**
>
> Thank you for your comments.
> Our training data, as well as other open-sourced R1-related datasets, are primarily focused on the mathematics domain.
> To further evaluate the generalization ability of R1-Share, we considered challenging out-of-domain (OOD) benchmarks like MMMU and AI2D.
> Specifically, MMMU covers multiple disciplines, and AI2D focuses on science diagrams. Both benchmarks differ significantly from the training data.
> We compared R1-Share with MM-Eureka, since our training data is a subset of that used for MM-Eureka. On MMMU, existing reasoning methods like MM-Eureka achieve an accuracy of 55.3\%, whereas R1-Share achieves 58.1\%, marking a 2.8-point improvement.
> For AI2D, R1-Share also achieves state-of-the-art performance, with R1-Share-32B reaching 86.2\% accuracy.
> These results demonstrate that R1-Share offers superior OOD generalization capabilities compared to GRPO.
>
> | Methods         | MMMU | AI2D |
> |-----------------|------|------|
> | MM-Eureka-7B    | 55.3 | 40.3 |
> | R1-Share-7B     | 58.1 | 42.9 |
> | MM-Eureka-32B   | 64.6 | 85.4 |
> | R1-Share-32B    | 70.1 | 86.2 |
>
> ---
>
> **W2: Apply ShareGRPO in NLP domain**
>
> This is a question well worth exploring.
> We conducted a preliminary exploration by applying ShareGRPO to LLM, and observed consistent performance improvements in this setting as well.
> We trained the Qwen2.5-7B-Instruct model using both ShareGRPO (w/o multi-modal transformations) and GRPO methods, with 5K samples from the Light-R1 dataset.
> The results show that ShareGRPO can also be applied to the NLP domain and achieves better performance than GRPO.
>
>
> | Methods                        | AIME'24 | GPQA |
> |--------------------------------|---------|------|
> | Qwen2.5-7B-Instruct (Baseline) | 12.0    | 36.4 |
> | w/ GRPO                        | 37.1    | 40.3 |
> | w/ ShareGRPO                   | 40.3    | 42.9 |
>
> ---
>
> **W3: Comparison between GRPO and DPO**
>
> Thank you for your comment.
> SeVa [1] is an insightful and relevant work which uses designed augmentation to the image input to generate negative data for DPO training, and we will cite it in the revised manuscript. GRPO is designed to incentivize long-chain reasoning capabilities without pre-generated reasoning annotations, while DPO training requires reasoning data to be generated in advance, making the training results highly dependent on the quality of the dataset.
>
> To delve deeper into this question, we include a comparison to further examine the reasoning capabilities incentivized by each method, by constructing DPO data based on the same underlying dataset.
> Using our 52K dataset, we use Qwen2.5-VL-72B generates long-chain steps to serve as the chosen data for DPO.
> And we prompt GPT-4o to generate reject responses using the method from SeVa [1] and STE [2].
> The results show that GRPO is more effective at incentivizing the model’s reasoning ability.
>
> | Methods | MathVista |
> |---------|-----------|
> |Baseline | 68.2      |
> | DPO     | 70.1      |
> | GRPO    | 72.8      |
> | ShareGRPO   | 75.4  |
>
>
> [1] Self-Supervised Visual Preference Alignment
>
> [2] Self-Taught Evaluators
>
> ----
>
> Thank you again for your review and for recognizing our work. We are encouraged by your feedback and will do our best to further refine this manuscript.
> If you feel your questions have been addressed, we would be grateful if you could kindly consider raising your score. Thank you!

---

> > ### Comment · Reviewer_MP3K · 2025-08-04
> >
> > Thank you for the author's response.
> >
> > Reply to W1: Indeed, the open-sourced R1-related datasets mainly focus on the mathematics domain. However, this paper uses accuracy reward, and applying GRPO on science-related multiple-choice datasets like SQA [Learn to Explain: Multimodal Reasoning via Thought Chains for Science Question Answering] and M$^3$CoT [M$^3$CoT: A Novel Benchmark for Multi-Domain Multi-step Multi-modal Chain-of-Thought] should not be difficult. Why not include these types of data in the training set? Are there any challenges when mixing data from different domains? I see that W2 performs quite well on the GPQA science questions.

---

> > > ### Author Response · Authors · 2025-08-05
> > >
> > > Thank you for your valuable comment! We’re glad to continue the discussion with you during this phase.
> > >
> > > -----
> > >
> > > > Why not include these types of data in the training set?
> > >
> > > To ensure a fair comparison in terms of data scale and source, and to follow prior reasoning work, we did not use these types of data. Instead, we focused on demonstrating the effectiveness of our method using similar data.
> > >
> > > > Are there any challenges when mixing data from different domains?
> > >
> > > Thank you for raising this concern. There are no challenges in mixing these datasets. Due to time constraints during the rebuttal period, we only sampled 5K additional examples from SQA and M3CoT and combined them with the original 52K samples used in R1-Share. The results show performance gains on both MMMU and AI2D, demonstrating that our method can effectively train on mixed-domain data.
> > >
> > > | Methods | MMMU | AI2D |
> > > |---------|-----------|------|
> > > | MM-Eureka |   55.3     |    84.1      |
> > > | Qwen2.5-VL-7B + GRPO    |   56.4    |   84.0   |
> > > | Qwen2.5-VL-7B + ShareGRPO   | 58.1  |   84.5   |
> > > | Qwen2.5-VL-7B + ShareGRPO + 5K additional data from SQA and M3CoT   | 59.1  | 85.4 |
> > >
> > > [1] Learn to Explain: Multimodal Reasoning via Thought Chains for Science Question Answering
> > >
> > > [2] M3CoT: A Novel Benchmark for Multi-Domain Multi-step Multi-modal Chain-of-Thought
> > >
> > > ----
> > >
> > > Thank you again for your professional review! We will incorporate our discussion into the revised paper to better present R1-Share. If you have any further questions or suggestions, please feel free to comment here.

---

### Official Review · Reviewer_EzA1 · 2025-06-26

**Clarity:** 2
**Significance:** 2
**Originality:** 2
**Rating:** 3
**Confidence:** 4

**Summary:**

The paper proposes a variant of GRPO applied to multimodal LLMs. The core of the method is to augment each question with different transformations to enrich the training data. The GRPO algorithm is then modified to incorporate two levels of normalization: one across all variants of the same question and another within each individual variant. The stated motivation is that this two-tiered normalization scheme allows for more precise credit assignment.

**Questions:**

- I would be interested to know the performance without the global shared advantage and with only the local shared advantage (but with the same data augmentation). In such a case, the method would revert to the normal GRPO, where the advantage is only normalized across the same question (specifically, the same variant of the same question). I suspect that simply increasing the number of samples per question variant would be sufficient for enhancing credit assignment, making the global advantage normalization unnecessary.
- Ideally, a learning curve should also be shown in the paper to understand whether the optimization has converged, especially for ablation runs.

**Ethical Concerns:**

["NO or VERY MINOR ethics concerns only"]

**Final Justification:**

The author addressed my concerns regarding computational cost and the similarities to GRPO, particularly clarifying the pairing of answers to all variants of the question during policy updates. As such, I have slightly increased my score.

Nonetheless, some issues remain:

1. Statistical significance: The author claims that their method is significant, but no statistical significance test is provided. Their main argument is that a common method A gives +x%, method B gives +y%, and y% is n times x%, implying that method B’s improvement is significant. However, this is not a valid statistical measure. Standard error and a t-test are required to establish statistical significance.
2. Limited novelty: The method is essentially a combination of data augmentation, a variant of group definition, and pairing of off-policy responses. In my opinion, none of these individual components are novel. A deeper analysis of how those off-policy pairs contribute to performance could yield novel insights, but the current submission falls short in this regard.

**Limitations:**

The authors addressed several limitations common in multimodal LLMs.

**Paper Formatting Concerns:**

None.

**Quality:**

2

**Strengths And Weaknesses:**

Strengths:

- The experiments show promising results, with the proposed model performing better than the baselines. The demonstrated scalability to a 32B model is also a notable strength.
- The method is straightforward and appears easy to implement.

Weaknesses:

- The major weakness is the limited novelty. In essence, it is a data augmentation technique. Both the "offline textual SCT" and "online multimodal SCT" could be performed in a purely offline manner; one could simply apply the online transformations prior to training to generate a large number of questions. The shared advantage computation is a straightforward combination resulting from two different group definitions and is closer to an engineering detail than a separate algorithm justifying a new name.
- The notation could be clearer and more rigorous. For example, the definitions of i, j, and k in A_{i,j,k} of equation (5) are not provided (though they can be inferred). The std function over a set of sets in equation (5) is not formally defined. In equation (6), the right-hand side should include a notation for k to clarify which original question variant Q_j belongs to. This leads to the more confusing equation in (7), which could be written more compactly without using cases.

---

> ### Author Rebuttal · Authors · 2025-07-31
>
> Dear reviewer EzA1,
>
> We appreciate your valuable comments and suggestions.
> We are glad to hear that you find our experimental results promising and our method straightforward and easy to implement.
> Below, we are going to address your concerns one by one to the best of our ability.
>
> ---
>
> **W1: Clarification on our motivation and contribution**
>
> Many thanks for raising this concern.
>
> > The major weakness is the limited novelty. In essence, it is a data augmentation technique. Both the "offline textual SCT" and "online multimodal SCT" could be performed in a purely offline manner; one could simply apply the online transformations prior to training to generate a large number of questions.
>
> We would like to kindly clarify that the key innovation of ShareGRPO is not simply the use of data transformation. We only use transformations to generate semantically consistent question variants and create shared responses for subsequent joint optimization.
> The core idea lies in leveraging shared and diverse responses for both self-updating ($\pi_{\theta}(o_i^{Q_j} \mid Q_k)$, where $j = k$) and cross-updating ($\pi_{\theta}(o_i^{Q_j} \mid Q_k)$, where $j \neq k$), each based on their respective conditional probabilities.
>
> In addition, we also conducted experiments to compare the results of data augmentation and ShareGRPO. The results indicate that the performance gains of ShareGRPO are not due to data augmentation, but rather to the mitigation of advantage vanishing through shared responses.
>
> | Methods    | GRPO | GRPO + image transformations | GRPO + language transformations | GRPO + image/language transformations | ShareGRPO |
> |-----------|-----------|--------|-------|------|------|
> | MathVista | 72.8      | 73.4   | 73.1  | 73.6 | 75.4 |
>
> > The shared advantage computation is a straightforward combination resulting from two different group definitions and is closer to an engineering detail than a separate algorithm justifying a new name.
>
> We respectfully clarify that the global shared advantage is also an innovation of this paper, specifically designed to facilitate shared advantage updates. It further enhances both the direction and magnitude of credit assignment to $\pi_{\theta}(o_i^{Q_j} \mid Q_k)$.
> Unlike GRPO, which computes the advantage within the set of responses generated for each individual question, ShareGRPO extends this to a hierarchical advantage calculation that incorporates both global and local advantages. The local advantage calculation differs from that in standard GRPO, while the global advantage refers to computing the advantage across all responses generated from all question variants. This specialized advantage calculation in ShareGRPO leads to improved performance, as shown in the table below.
>
> | Methods    | Local advantage | Global advantage | MathVista
> |-----------|:-----------:|:--------:|:-------:|
> | ShareGRPO | ✔️    |    |  74.1 |
> | ShareGRPO |     | ✔️   | 74.8  |
> | ShareGRPO | ✔️    | ✔️   | 75.4  |
>
> Lastly, we summarize the contributions of this work as follows to highlight its uniqueness: We propose Share-GRPO, the first method to introduce information sharing into MLLM reinforcement learning. It shares diverse reasoning trajectories across an expanded question space to alleviate sparse reward and advantage vanishing issues. We further design a hierarchical advantage estimation mechanism that shares reward information both across and within question variants, enabling more accurate and robust estimation. Finally, extensive experiments on six MLLM reasoning benchmarks validate the effectiveness of our approach.
>
> ---
>
> **W2: Refine Equations (5), (6), and (7)**
>
> Thank you for taking the time to carefully review our work and point out these issues!
> We agree that the notation could be clearer and more rigorous. We originally added the index $k$ in Equations (5) and (6) because, in Equation (8), we need to distinguish the calculations of $A$ for different policies $\pi_{\theta}(\cdot \mid Q_k)$. Below, we adopt a different formulation of the equations and provide a detailed explanation of the formulas.
>
> The indices $k$ and $j$ are primarily used in Equation (8) for the conditional probability $\pi_{\theta}(o_i^{Q_j} \mid Q_k)$. Their purpose is to distinguish between the response $o_i$ generated from $Q_j$ and the question variant $Q_k$ on which the conditional probability is evaluated. This allows us to differentiate between self-updating (when $j = k$) and cross-updating (when $j \neq k$) scenarios in the computation.
>
> For $\pi_{\theta}(o_i^{Q_j} \mid Q_k)$, when $k = j$, all responses $o_i$ are generated from $Q_j$ and are used to update the same question $Q_k$. In this case, we apply both global and local advantage estimation, as these responses are directly associated with their original variant.
>
> $$
> A_{i, j}^{\text{hier}} = A_{i, j}^{\text{global}} + A_{i, j}^{\text{local}}
> $$
>
> For $\pi_{\theta}(o_i^{Q_j} \mid Q_k)$, when $k \neq j$, the response $o_i$ is generated from $Q_j$ but used to update a different variant $Q_k$. Here, we adopt only the global advantage estimation, since the response is being shared across variants.
>
> $$
> A_{i, j}^{\text{hier}} = A_{i, j}^{\text{global}}
> $$
>
>
> For global advantage computation, we estimate the advantage from a global perspective, where the relative advantage is computed using the rewards obtained from all question variants.
>
> $$
> A_{i, j}^{\text{global}} = \frac{R_i^{Q_j} -\text{mean}\left(\{{[\{r_{1}^{Q_1}, ..., r_{n}^{Q_1}}\}], ..., [\{r_{1}^{Q_m}, ..., r_{n}^{Q_m}\}\}]\right)}{\text{std}\left(\{[\{r_{1}^{Q_1}, ..., r_{n}^{Q_1}\}], ..., [\{r_{1}^{Q_m}, ..., r_{n}^{Q_m}\}\}]\right)}.
> $$
>
> For local advantage computation, we also estimate the advantage at a local level, where the relative advantage is computed within the responses generated from each individual question variant Qj ∈ Q. Specifically, for each question variant Qj, the local advantage is estimated as follows:
>
> $$
> A_{i, j}^{\text{local}} =
> \frac{
>     R_i^{Q_j} - \mathrm{mean}\left(\{ r_1^{Q_j}, ..., r_n^{Q_j} \}\right)
> }{
>     \mathrm{std}\left(\{ r_1^{Q_j}, ..., r_n^{Q_j} \}\right)
> }
> $$
>
> Thank you for you suggestions. We have revised and clarified these expressions in the updated manuscript.
>
> ----
>
> **Q1: The results without global shared advantage and the clarification of advantage computation**
> Thank you for your question.
>
> > I would be interested to know the performance without the global shared advantage and with only the local shared advantage (but with the same data augmentation).
>
> We provide results without the global shared advantage, which show a slight decrease in performance. This is because the global shared advantage offers higher confidence in credit assignment.
>
> | Methods    | Local advantage | Global advantage | MathVista
> |-----------|:-----------:|:--------:|:-------:|
> | ShareGRPO | ✔️    |    |  74.1 |
> | ShareGRPO |     | ✔️   | 74.8  |
> | ShareGRPO | ✔️    | ✔️   | 75.4  |
>
> > In such a case, the method would revert to the normal GRPO, where the advantage is only normalized across the same question (specifically, the same variant of the same question).
>
> We believe that there may be some misunderstanding.
> ShareGRPO, even when using only local shared advantage and without the global shared advantage, **does not revert to normal GRPO**.
> We would like to kindly highlight that the essential innovation of ShareGRPO is not simply the introduction of transformation or the use of a global shared advantage.
> The core idea lies in leveraging shared and diverse responses for both self-updating ($\pi_{\theta}(o_i^{Q_j} \mid Q_k)$, where $j = k$) and cross-updating ($\pi_{\theta}(o_i^{Q_j} \mid Q_k)$, where $j \neq k$), each based on their respective conditional probabilities.
> The semantically consistent transformation is primarily a means to create a shared and diverse response pool, while the global shared advantage further enhances both the direction and magnitude of credit assignment to $\pi_{\theta}(o_i^{Q_j} \mid Q_k)$.
>
>
> > I suspect that simply increasing the number of samples per question variant would be sufficient for enhancing credit assignment
>
> We agree that simply increasing the number of samples per question would enhance credit assignment.
> However, as shown in Table 5, simply increasing the number of rollouts leads to more homogeneous responses and quickly reaches a performance ceiling. For example, when the GRPO rollout number increases from 12 to 24, the accuracy on MathVista improves by only 0.2\%, while significantly increasing computational burden.
> Therefore, it is more effective and efficient to increase the number of question variants and adopt shared response updating as used in ShareGRPO.
>
> **Q2: Inclusion of learning curve**
>
> Thank you for your suggestion!
> We agree that providing learning curves is valuable for analyzing optimization and convergence.
> Accordingly, we have included learning curves in Figure 1(b) of the main paper.
> While we would like to provide additional learning curves in the rebuttal, technical restrictions from NeurIPS prevent us from submitting figures at this stage.
> We will include more learning curves in the revised version, such as reward curve, and response length curve.
>
> ---
>
> Thank you again for your insightful and professional comment, which made our work more complete and solid!
> If there are any further questions, please let us know.
> If you feel all questions have been addressed, we would be grateful if you could kindly consider re-rating our work. Looking forward to hearing back from you!

---

> > ### Comment · Reviewer_EzA1 · 2025-08-04
> >
> > Thank you for the response.
> >
> > > We would like to kindly clarify that the key innovation of ShareGRPO is not simply the use of data transformation.
> >
> > The results of `GRPO + image/language transformations` yield a performance of 73.6, which is very close to 75.4—this makes me question the statistical significance of the method beyond simple data augmentation. Could the authors comment on the statistical significance of this difference?
> >
> > > We believe that there may be some misunderstanding. ShareGRPO, even when using only local shared advantage and without the global shared advantage, does not revert to normal GRPO.
> >
> > Thank you for the clarification. I now see that the response to different variants is also included in the update. This could be clarified better in Section 3.2.3—for example, by stating: “We pair all combinations of sampled responses to variants of the same question in our update.” However, this leads to two important questions:
> >
> > 1. Would this pairing lead to a large increase in computational cost? It is quadratic in the number of variants per question. This large increase in computational cost seems to yield only a marginal benefit (i.e., from 73.6 to 75.4).
> > 2. In PPO or GRPO, if the importance ratio exceeds the clipped threshold, the update is essentially disabled. Pairing sampled responses to variants of the same question would result in off-policy samples and hence a ratio far from 1, effectively disabling all updates when j ≠ k. Is there any analysis on how off-policy these samples are?

---

> > > ### Author Response · Authors · 2025-08-05
> > >
> > > We are glad to hear from you.
> > >
> > > > The results of GRPO + image/language transformations yield a performance of 73.6, which is very close to 75.4—this makes me question the statistical significance of the method beyond simple data augmentation. Could the authors comment on the statistical significance of this difference?
> > >
> > > Thank you! We respectfully clarify that the improvement from 73.6 to 75.4 is significant, for the following reasons:
> > >
> > > **First**, the baseline performance of Qwen2.5-VL with GRPO is 72.8%. Applying simple image/language transformations increases it to 73.6%, a modest 0.8% gain. In contrast, R1-Share improves over Qwen2.5-VL + GRPO by 2.6% and exceeds GRPO with transformations by 1.8%, demonstrating that our improvement is substantial and well beyond what simple data transformations achieve.
> > >
> > > |Models|MathVista|
> > > |-|:-:|
> > > |GRPO|72.8|
> > > |GRPO + image/language transformations|73.6 (**+0.8%**)|
> > > |Our ShareGRPO|75.4  (**+2.6%**)|
> > >
> > > **Second**, we believe this performance gain is highly significant within this field.
> > > 1) MathVista accuracy is already high, making further gains difficult. For example, even the powerful GPT-o1 achieves only 73.9%, below our R1-Share-7B’s 75.4%.
> > > 2) ShareGRPO achieves a larger improvement than other methods. For instance, QVQ, based on Qwen2-VL-72B, gains only 0.9% on MathVista, and adding DAPO’s Dynamic Sampling to GRPO yields just 0.8%. In contrast, our model improves over GRPO by 2.6%, and over GRPO with transformations by 1.8%, demonstrating a substantial performance gain.
> > >
> > > |Models|MathVista|
> > > |-|-|
> > > |Qwen2-VL-72B|70.5|
> > > |QVQ-72B (Qwen + enhanced reasoning capabilities)|71.4 (**+0.9%**)|
> > >
> > > |Models|MathVista|
> > > |-|-|
> > > |Qwen2.5-VL-7B  + GRPO|72.8|
> > > |Qwen2.5-VL-7B  + GRPO + DAPO's Dynamic Sampling|73.6 (**+0.8%**)|
> > >
> > > |Models|MathVista|
> > > |-|-|
> > > |GRPO + Transformation|73.6|
> > > |ShareGRPO|75.4 (**+1.8%**)|
> > >
> > > > 1. Would this pairing lead to a large increase in computational cost? It is quadratic in the number of variants per question. This large increase in computational cost seems to yield only a marginal benefit (i.e., from 73.6 to 75.4).
> > >
> > > Thank you for raising this concern! We’d like to clarify that the performance gain is significant (please refer to our response above), and the associated computational overhead is small and acceptable.
> > >
> > > For computational cost, ShareGRPO introduces a moderate overhead of about 20% compared with GRPO, which we consider acceptable. This is because we keep the total number of rollouts constant: GRPO generates 12 rollouts for one query, while ShareGRPO generates 6 rollouts each for two variants, or 4 rollouts each for three variants—still totaling 12. Thus, overall rollout cost and advantage computation remain the same.
> > > The additional computation primarily arises during cross-updating in the policy optimization phase.
> > > Given the performance gains, we believe this 20% runtime increase is a reasonable trade-off.
> > >
> > > > 2. In PPO or GRPO, if the importance ratio exceeds the clipped threshold, the update is essentially disabled. Pairing sampled responses to variants of the same question would result in off-policy samples and hence a ratio far from 1, effectively disabling all updates when j ≠ k. Is there any analysis on how off-policy these samples are?
> > >
> > > Thank you. It is a very interesting point worth discussing.
> > > In our experiments, the proportion of samples with importance ratios exceeding the upper clipping threshold was 0.001 in GRPO and 0.003 in ShareGRPO. We believe this does not affect training, for the following reasons:
> > > 1) Although ShareGRPO responses are conditioned on different question variants, they are generated by the same model (i.e., the current policy), not a different one. This limits off-policy deviation and keeps importance ratios close to 1.
> > > 2) While ShareGRPO shows a slight increase in the proportion of samples with importance ratios exceeding the upper clipping threshold (i.e., 0.001 to 0.003), the overall proportion remains low, and we do not consider it a cause for concern.
> > > 3) Importantly, our model’s performance improves: GRPO achieves 72.8% accuracy, while ShareGRPO reaches 75.4%, indicating that this phenomenon does not affect learning.
> > > 4) Furthermore, prior work such as DAPO [1] has shown that learning from high importance ratio samples can enhance exploration and improve training. This has inspired techniques like clip-higher [1], which allow for larger importance sampling ratios. Our findings are consistent with these results, suggesting that the modest increase in importance sampling ratios in ShareGRPO may positively impact both exploration and performance.
> > >
> > > [1] DAPO: An Open-Source LLM Reinforcement Learning System at Scale
> > >
> > > ---
> > >
> > > We appreciate your professional review and sincerely believe that your suggestions have helped improve the quality of our paper.
> > > We will incorporate our discussion into the revised version. Please feel free to reach out if you have any further questions or suggestions.

---

> > > > ### Comment · Reviewer_EzA1 · 2025-08-07
> > > >
> > > > Thanks for the response. I have one last question:
> > > >
> > > > > This is because we keep the total number of rollouts constant: GRPO generates 12 rollouts for one query, while ShareGRPO generates 6 rollouts each for two variants, or 4 rollouts each for three variants—still totaling 12. Thus, overall rollout cost and advantage computation remain the same.
> > > >
> > > > If I understand correctly, in the case of 4 rollouts for each of the three variants, the batch size during policy training would be calculated as 4×3×3=36 (i.e., 12 rollouts in total, each paired with three variants). This implies that the training cost increases roughly threefold. Since small mini-batches are typically required to fit into GPU memory, the benefit of larger batch sizes would be limited—so the computational cost should scale approximately linearly.
> > > >
> > > > However, the authors commented that the additional computational cost is only around 20%, which seems inconsistent. Does this imply that policy training accounts for only a small portion of the overall training time? From my experience with RLVR on 1.5B and 7B models, policy training typically accounts for roughly half of the total training time, with most of the remaining time spent on rollout generation. The computational cost of rollout generation should indeed remain the same, assuming the number of rollouts is unchanged. A detailed breakdown of the computation time across different components would be very helpful for clarification.

---

> ### Author Response · Authors · 2025-08-07
>
> Thank you for your response!
>
> we believe your understanding is correct! We apologize for any confusion caused. The 20% increase we mentioned refers specifically to the case where the number of questions is 2 and the number of rollouts is 6. We will clarify this point in the revised paper. In fact, our default setting throughout the paper is with two questions, as we found this configuration already yields strong results (if you are interested in more details for this part, please refer to our responses to Reviewer Amky’s W1 and Reviewer v7hu’s W5(b)). This setting also strikes a good balance between performance and training efficiency.
>
> For the detailed computation time analysis, we used the EasyR1 framework for training and monitored the process using wandb. The table below presents the computation time for each component within a single RL step, arranged in the order of execution, with the total time per step shown at the end. As shown, ShareGRPO only increases the computation time per step by 20%.
>
> This phenomenon can be explained by the fact that, in each training phase, parallel computation (which is increased by methods such as ShareGRPO) only accounts for part of the total time per component. Other substantial fixed or distributed overheads—such as parameter synchronization, communication latency, buffer initialization, and scheduling—remain largely unchanged and unavoidable. As a result, the overall increase in computation time is much less than the theoretical complexity might suggest.
>
> | Components in RL | Rollout | Recompute log_probs | Recompute ref_probs| Compute advantage | Compute reward |  Update actor |  Total time each step |
> | -- | -- | -- | -- | -- | -- | -- | -- |
> | time (s) in GRPO | 80 | 117 | 117 | 0.16 | 6 | 365 | 685.16 |
> | time (s) in ShareGRPO | 80 | 146 | 146 | 0.34 | 9 | 447 | 828.34 (+20.89%)|
>
>
> ------
>
> Thank you again for your valuable and professional comments! We hope that our response has addressed your concerns. If so, we would greatly appreciate it if you could kindly consider re-rating our work. If you have any further questions or suggestions, please feel free to comment here.

---

> ### Author Response · Authors · 2025-08-09
> **Kindly Confirm Concerns Addressed and Any Further Suggestions**
>
> Dear Reviewer EzA1,
>
> We hope this message finds you well. As the discussion period in nearing its end with less than 8 hours remaining,
> we are writing to kindly check whether we have addressed all your concerns satisfactorily.
> If there are any additional points or feedback you'd like us to consider, please let us know.
> Your insights are invaluable to us, and we will incorporate your suggestions into the revised version. Lastly, we will remain actively engaged in the discussion until the end of the rebuttal period to address any remaining issues.
>
> Thank you very much for your review!

---

> > ### Comment · Reviewer_EzA1 · 2025-08-09
> >
> > Thank you for the response, which was helpful for my final assessment.

---

> > > ### Author Response · Authors · 2025-08-09
> > >
> > > Thank you for your reply! We’re glad to know that our responses were helpful for your final assessment, and we truly appreciate your time and professional review.

---

### Official Review · Reviewer_v7hu · 2025-06-27

**Clarity:** 2
**Significance:** 2
**Originality:** 3
**Rating:** 5
**Confidence:** 4

**Summary:**

This paper proposes a Share-GRPO mechanism to train multimodal LLMs. Share-GRPO first augments the original problem into multiple variants, then employs the policy model to generate multiple candidate reasoning responses for all augmented variants. When updating the policy, Share-GRPO introduces an advantage-sharing mechanism to provide a rich reward signal to the policy model, alleviating the reward sparsity issues in standard GRPO. Extensive experiments demonstrate the effectiveness of the proposed Share-GRPO.

**Questions:**

1. In Equation.(7), what does $\pi_\theta(o_i^{Q_j}|Q_k)$ mean? The notation $o_i^{Q_j}$ is already defined as "response $o_i$ generated by the policy model conditioned on problem variant $Q_j$." However, the conditional notation $(\cdot | Q_k)$ is confusing to me. I would appreciate it if the authors could clarify this further.
2. The proposed Share-GRPO framework also appears to be seamlessly applicable to the text-only reasoning domain. I wonder whether the authors have conducted experiments using text-only LLMs on challenging mathematical benchmarks such as AIME 2024, GPQA, etc.

**Ethical Concerns:**

["NO or VERY MINOR ethics concerns only"]

**Final Justification:**

The authors’ rebuttal has effectively addressed my concerns regarding training robustness and the advantage vanishing issue in Share-GRPO, so I have decided to raise my score. However, due to some intrinsic limitations of Share-GRPO, such as the additional computational cost introduced by augmentation, I prefer to give a score of 4.5, which corresponds to a weak accept by NeurIPS 2024 standards.

**Limitations:**

See weaknesses.

**Paper Formatting Concerns:**

No major formatting concerns.

**Quality:**

3

**Strengths And Weaknesses:**

Strengths:
1. The idea of Share-GRPO, which augmenting the original problem to provide rich reward signal to alleviate reward sparsing issue, is convincing and reasonable.
2. The experiments presented in the paper are comprehensive and solid.
3. The paper is well-structured and mostly clearly written.

Weaknesses:
1. As presented in Limitations, the data augmentation done by generative models such as GPT-4o may introduce noise into augmented problems.
2. In Figure.1(b), although the valid advantage ratio curve of Share-GRPO is slightly higher than the standard GRPO's, their trends are almost the same, and their values are only slightly different from about step 20, finally becoming almost indistinguishable by step 70. I don't think Share-GRPO truly solve the advantage vanishing issue based on this figure.
3. Ablation studies and experiments in Tables 4 and 5 are only conducted on one benchmark "MathVista". It would be more convincing if the authors provide more results such as the average performance in Table.1.
4. In Equation.(5) and (6), the subscript "k" does not appear in the RHS of the equation. Its meaning here is unclear to me.
5. In Table.4, Share-GRPO achieves suprior performance only when setting $m=2$, which means only 1 augmentation is conducted for each problem, with the other being the original problem in my understanding. I have 2 concerns about this observation:

(a). In Section 3.2.1, the authors introduce several augmentation approaches. I wonder which one was ultimately chosen for the experiments. If it was selected randomly, could the reproducibility of Share-GRPO be weakened, since different augmentations may heavily affect the data samples?

(b). Why does increasing *m* from 1 to 2 lead to a significant performance gain (72.8 → 75.4 on MathVista), while increasing *m* from 2 to 4 yields only a minimal improvement (75.4 → 75.9 on MathVista)? I’m interested in this discrepancy, but the authors do not discuss it.

---

> ### Author Rebuttal · Authors · 2025-07-31
>
> Dear reviewer v7hu,
>
> Many thanks for your valuable comments and questions, which help us a lot to improve our work.
> We are glad that you find ShareGRPO’s richer reward signals convincing, and that you consider our experiments comprehensive and the paper well-structured.
> We are going to try our best to address your questions one by one as follows.
>
> ---
>
> **W1: Data quality concerns regarding GPT-4o-based problem augmentation**
>
> Thank you for raising this concern.
> We have taken several measures to ensure the quality of the augmented problems, as detailed below, and we will include these in the revised paper:
> 1) We experimented with various prompts to ensure stable and high-quality outputs.
> 2) We developed rule-based scripts to automatically filter out incomplete generations, ensuring that the retained sentences are complete and that the options remain consistent with the original questions.
> 3) We manually reviewed a large number of cases to further ensure the quality and correctness of the outputs.
> 4) We also conducted GRPO training using only one newly generated question variant. The training process was stable and achieved a slight performance improvement as shown below, which empirically demonstrates the quality of the generated questions.
>
> | Models    | one original question | one new question  |
> |-----------|-----------|--------|
> | GRPO      | 72.8      | 72.9   |
>
> We will include these discussion in the revision.
>
> ---
>
> **W2: Discussion about Figure.1(b) and advantage vanishing issue**
>
> Thank you for raising this concern.
> We would like to kindly clarify that there are many factors that lead to the advantage vanishing problem. For example, (1) Memorization and reward hacking: MLLMs may memorize the final answer due to data leakage and can produce the correct final answer even when intermediate reasoning steps are incorrect. Please refer to our response to the reviewer Amky's W1 for further details about memorization issues; (2) Trapped in low quality questions: For certain low-quality questions, the MLLM is unable to produce correct answers, leading to all rollouts being incorrect; and (3) Genuine reasoning ability: The model may have truly learned to solve the problem through reasoning, leading to consistently correct answers.
>
> ShareGRPO is primarily designed to address issues (1) and (2) by generating shared and diverse responses through semantically consistent transformations. These responses are then used for both self-updating ($\pi_{\theta}(o_i^{Q_j} \mid Q_k)$, where $j = k$) and cross-updating ($\pi_{\theta}(o_i^{Q_j} \mid Q_k)$, where $j \neq k$), based on their respective conditional probabilities.
>
> As training progresses, the reasoning ability of MLLMs steadily improves, enabling them to genuinely solve problems, even when presented with semantically varied questions, and to develop true reasoning skills.
> This inevitably leads to more cases where the model answers all variants correctly, thus resulting in training convergence and a decrease in the valid advantage ratio.
> To address issue (3), dynamic sampling from DAPO [1] is specifically designed for such scenarios. As shown in Table 3, our results demonstrate the complementarity and compatibility of ShareGRPO with dynamic sampling.
>
> [1] DAPO: An Open-Source LLM Reinforcement Learning System at Scale
>
> ---
>
> **W3: Average performance in Tab.4 and Tab.5**
>
> This is a good question!
> We have provided the average performance results below. We believe including these results makes our findings more convincing. Thank you!
>
> | Models    | MathVista | MMStar | MMMU  | MathVerse | MathVision | AI2D | AVG  |
> |-----------|-----------|--------|-------|------------|-------------|------|------|
> | GRPO      | 72.8      | 65.4   | 56.4  | 50.7       | 26.7        | 84.0 | 59.3 |
> | ShareGRPO | 75.4      | 67.0   | 58.1  | 52.8       | 29.5        | 84.5 | 61.2 |
>
>
> | Method     | Question | average |
> |------------|----------|---------|
> | GRPO       | 1        | 59.3    |
> | ShareGRPO  | 2        | 61.2    |
> | ShareGRPO  | 3        | 61.4    |
> | ShareGRPO  | 4        | 61.8    |
>
>
> | Method     | Sampling | MathVista |
> |------------|----------|-----------|
> | GRPO       | 6        | 58.7      |
> | GRPO       | 12       | 59.3      |
> | GRPO       | 24       | 59.7      |
> | ShareGRPO  | (3+3)    | 60.7      |
> | ShareGRPO  | (6+6)    | 61.2      |
>
> ---
>
> **W4: Clarification on the notation in Eq.5 and Eq.6**
>
> Thank you for pointing out the confusion regarding the subscript "k" in Eq. (5) and (6).
> We acknowledge that the notation could be more clearly defined, and we will revise the equations and provide additional explanations in the revised version to ensure clarity.
> **Due to character limitations**, we have provided a detailed explanation in our response to **Reviewer EzA1’s W2**. Please kindly refer to that section for further details.
>
> ---
>
> **W5 (a): Clarification on the reproducibility of ShareGRPO**
>
> Thank you for raising this concern.
> We fixed the random seed during training and conducted five independent runs.
> The results show that the performance fluctuation is minor. We believe this level of variation is acceptable, as similar fluctuations are also observed in GRPO, due to the inherent randomness in reinforcement learning.
>
> | Methods                   | GRPO           | ShareGRPO       |
> |---------------------------|----------------|------------------|
> | Error bars on MathVista   | 72.8 ± 0.3     | 75.4 ± 0.39      |
>
>
> ---
>
> **W5 (b): Explanations on the Performance Gains with Increasing the Number of Question variants $m$**
>
> Thank you for raising this concern!
> Based on Table 3 and Table 4, we observe that increasing the number of question variants from $m=1$ to $m=2$ leads to a significant performance improvement (72.8\% → 75.4\% on MathVista).
> This improvement is primarily due to our use of semantically consistent transformations and shared response optimizations, which effectively address the advantage vanishing issue.
> However, the gain is smaller when increasing $m$ from 2 to 4. This is because, with two variants, the MLLM is already able to generate both positive and negative rollouts, which sufficiently addresses the sparse reward issue. As a result, further increasing $m$ provides diminishing returns. Therefore, to balance efficiency and performance, we use two question variants as the default setting.
>
> ---
>
> **Q1: Explanation of Eq.7**
>
> Thank you for your comment!
> We first clarify the meaning of $\pi_{\theta}(\cdot \mid Q)$ in GRPO.
> In GRPO, for a given question $Q$, we first roll out $n$ responses and, for each response, compute both its advantage $A$ and the conditional probability $\pi_{\theta}(\cdot \mid Q)$ for each token in the response. Here, $\pi_{\theta}(\cdot \mid Q)$ denotes the parameterized policy (with parameters $\theta$) that outputs a probability distribution over possible tokens conditioned on $Q$.
> If the advantage is positive, the probability of the corresponding tokens is increased; if the advantage is negative, the probability of these tokens is suppressed.
>
> > In Equation.(7), what does $\pi_{\theta}({o_i^{Q_j}}  \mid {Q_k})$ mean?
>
> In ShareGRPO, we extend this framework by introducing multiple question variants [$\{Q_1, Q_2, \dots, Q_m\}$]. For each variant $Q_j$, we generate a set of responses $\mathbf{O}=[[\{\{o_{1}^{Q_1}, ..., o_{n}^{Q_1}\}], ..., [\{o_{1}^{Q_m}, ... o_{n}^{Q_m}\}\}]]$. Each response $o_i^{Q_j}$, where j belongs to m, is thus explicitly generated by the policy model conditioned on $Q_j$.
>
> The notation $\pi_{\theta}(o_i^{Q_j} \mid Q_k)$ in Eq. (7) represents the probability (under the current policy) of generating the entire response $o_i^{Q_j}$, but conditioned on a possibly different question variant $Q_k$.
>
> 1) When $j = k$, this is the “self-updating” case, where we update the parameters using the response under its own generating context.
>
> 2) When $j \neq k$, this is the “cross-updating” case, where we reuse a response generated from one question variant and update its probability under a different variant.
>
> This design allows ShareGRPO to perform cross-variant updates, promoting information sharing and robustness across semantically consistent but syntactically different variants.
>
> We appreciate your feedback and will further clarify this notation in the revised manuscript.
>
> ---
>
> **Q2: Apply ShareGRPO to text-only LLM**
>
> Thank you! This is certainly worth exploring further.
> We didn't conduct experiments with text-only LLMs before, as Share-GRPO was originally designed for MLLMs, incorporating features such as multi-modal image transformations.
> However, we agree that extending Share-GRPO to text-only reasoning tasks is a promising direction for future work.
> Hence, we conducted a preliminary exploration by applying ShareGRPO to pure text tasks, and observed consistent performance improvements in this setting as well.
> We trained the Qwen2.5-7B-Instruct model using both ShareGRPO (w/o multi-modal transformations) and GRPO methods, with 5K samples from the Light-R1 dataset.
> The results show that ShareGRPO achieves better results than GRPO.
>
>
> | Methods                          | AIME'24 | GPQA |
> |----------------------------------|---------|------|
> | Qwen2.5-7B-Instruct (Baseline)   | 12.0    | 36.4 |
> | w/ GRPO                          | 37.1    | 40.3 |
> | w/ ShareGRPO                     | 40.3    | 42.9 |
>
>
> ---
>
> Lastly, thank you very much for your constructive feedback and suggestions. We will incorporate these points into the revised paper. If you have any further questions, please feel free to let us know. We will be available throughout the rebuttal period.

---

> > ### Comment · Reviewer_v7hu · 2025-08-02
> >
> > Thank you for your detailed responses! Most of my concerns have been well addressed. However, a few issues remain:
> >
> > - W2: Are there any methods to quantify the proportion of these three factors? Providing their relative proportions and tracking their trends throughout the training process would strengthen the argument.
> > - W5(a): My question primarily concerns whether different random augmentations lead to significantly different final performance. Therefore, I think the experiments in the rebuttal should use varying random seeds, rather than a fixed one, to reflect this robustness.

---

> ### Author Response · Authors · 2025-08-05
>
> Thank you for your comment! We are glad to engage with you during the discussion phase.
>
> -------
>
> > W2: Are there any methods to quantify the proportion of these three factors? Providing their relative proportions and tracking their trends throughout the training process would strengthen the argument.
>
> This is a good suggestion! We collected 50 samples from our training data, and manually categorized them into the four types: (1) Questions with memorization and reward hacking issues, (2) Low-quality questions, (3) Under-Learned questions, (4) Well-learned questions. Among these, factors (1), (2), and (4) lead to advantage vanishing, while factor (3) represents samples the model is learning, which do not exhibit advantage vanishing.
>
> The definitions of these three categories are as follows:
> 1) Questions with memorization and reward hacking issues: The questions produce all correct final answers during rollouts, while its reasoning process contains clear intermediate errors, resulting in advantage vanishing.
> 2) Low-quality questions: The questions include minor issues such as typos or mixed-language usage. While these flaws do not hinder comprehension, the base model fails to answer them correctly in all rollouts, resulting in advantage vanishing.
> 3) Under-Learned questions: The questions produce both positive and negative samples during rollout.
> 4) Well-learned questions: The questions produce all-correct reasoning and final answer, resulting in advantage vanishing, where we believe this should be fine as the model can well solve these questions.
>
> We measure how the proportion of each category changes along the training to reveal that how ShareGRPO mitigates advantage vanishing issues.
> Specifically, to analyze the proportions and trends of advantage vanishing caused by these factors (i.e., (1), (2) and (4)) during training, we evaluated intermediate checkpoint models at training steps 25, 50, 75, and 100. Each model was tested on the same set of questions, and we manually reviewed the responses at each stage. This allowed us to track how the impact of each factor evolved over the training. The table below presents the proportion of advantage vanishing for each factor at different training stages.
>
> During training, the proportion of (1) memorization and (2) low-quality input cases steadily declined, while the proportion of (4) well-learned questions improved, demonstrating the effectiveness of our ShareGRPO and reinforcing our argument.
> Compared to GRPO, ShareGRPO more effectively mitigates the advantage degradation caused by issues (1) memorization and (2) low quality questions, ultimately leading to a higher proportion of correct reasoning and final answers and better accuracy.
>
> | Model after training (GRPO) | 1) Questions with memorization and reward hacking issues ↓ | 2) Low-quality questions ↓ | 3) Under-Learned questions | 4) Well-learned questions ↑ |
> |--|---|------|-----|----|
> | step 0    |  18%  | 14% | 68% | 0% |
> | step 25    |  16%  | 12% | 62% | 6% |
> | step 50   |  14%  | 10% | 58% | 18% |
> | step 75    |  14%  | 8% | 52% | 26% |
> | step 100  |  10%  | 8% | 48% | 34% |
>
> | Model after training (ShareGRPO)   | 1) Questions with memorization and reward hacking issues ↓ | 2) Low-quality questions ↓ | 3) Under-Learned questions | 4) Well-learned questions ↑ |
> |---|----|--|---|--|
> | step 0  | 12%  |  8% | 80% | 0% |
> | step 25     | 10%  |  6% | 72% | 12%|
> | step 50    | 6% |  4% | 64% | 26% |
> | step 75    | 6% |  4% | 58% | 32% |
> | step 100  | 4% |  2% | 52% | 42% |
>
> ------
>
> > W5(a): My question primarily concerns whether different random augmentations lead to significantly different final performance. Therefore, I think the experiments in the rebuttal should use varying random seeds, rather than a fixed one, to reflect this robustness.
>
> Thank you very much for this question!
> We conducted three runs with different random seeds. The experimental results show that our method is reproducible and robust. As training progresses, the models consistently converge to similar performance levels, demonstrating stable behavior across different random augmentations.
>
> | Method   | GRPO (Fixed Seed)     | ShareGRPO (Fixed Seed) | GRPO (Varying Seeds)  | ShareGRPO (Varying Seeds)  |
> |--------------------------|-----------------------|-------------------------|----------------------------|----------------------------|
> | Error bars on MathVista | 72.8 ± 0.3            | 75.4 ± 0.39          | 72.8 ± 0.37         | 75.4 ± 0.44           |
>
> -----
>
> We will include these discussions in the revised version. Thank you again for your constructive response!

---

> > ### Comment · Reviewer_v7hu · 2025-08-07
> >
> > Thank you for the response. These statistics make the claim more convincing to me.
> >
> > However, there appear to be some inconsistencies between Figure 1 and the statistics in the provided table. In Figure 1, the valid advantage ratios (1 - %Well-Learned Questions) for Share-GRPO and GRPO become nearly indistinguishable around step 70. Yet in the table, the differences remain notable at step 75 (68% vs. 74%) and step 100 (58% vs. 66%), which warrants further clarification.

---

> > > ### Author Response · Authors · 2025-08-07
> > >
> > > Thank you for you comment!
> > >
> > > ----
> > >
> > > We would like to kindly clarify that the valid advantage ratios is [**%Under-Learned questions** or **1 - (%Questions with memorization and reward hacking issues + %Low-quality questions + %Well-learned questions)**]. The reason is questions with memorization and reward hacking, as well as low-quality questions, can also lead to vanishing advantages. In contrast, under-learned examples represent samples that are still being learned, which correspond to valid advantage.
> > >
> > > Accordingly, the valid advantage ratios for GRPO and ShareGRPO, as shown in the table above and in Figure 1, are as follows:
> > >
> > > | Valid advantage ratio | Step75 in Fig 1 | Step75 in Table above | Step 100 in Fig 1| Step 100 in Table above |
> > > |-|-|-|-|-|
> > > | GRPO | 51 | 52 | 50 | 48 |
> > > | ShareGRPO | 53 (+2%) | 58 (+6%) | 55 (+5%) | 52(+4%) |
> > >
> > > We believe this variation is reasonable, as our statistical analysis uses fixed question sets, while the training data changes at each step. This naturally results in slight discrepancies. However, more importantly, the consistent trend in valid advantage across both the table and Figure 1 can reinforces the reliability of the tabulated results.
> > >
> > > -----
> > >
> > > Many thanks for your response!

---

> > > > ### Comment · Reviewer_v7hu · 2025-08-07
> > > >
> > > > I appreciate all the clarifications provided, which have addressed my main concerns. I will adjust my score accordingly. Please ensure that the notations in the equations and the analysis of advantage vanishing are updated in the final version.

---

> > > > > ### Author Response · Authors · 2025-08-07
> > > > > **Many thanks for your professional review!**
> > > > >
> > > > > Thank you again for your review! In the revised version, we will ensure that the notations in the equations and the analysis of advantage vanishing are clearly included. We sincerely believe that your suggestions have significantly improved the quality of our paper. Thank you once again.

---

### Official Review · Reviewer_Amky · 2025-07-02

**Clarity:** 4
**Significance:** 3
**Originality:** 3
**Rating:** 4
**Confidence:** 4

**Summary:**

This work proposes Share-GRPO, a RL methodology for VLLM that expands the input question space by (1) use LLM to paraphrase the textual questions (2) transform the input image by pre-defined operations (e.g., rotation, noise injection). The author argues that the proposed transformation techniques introduce diversity and elicit diversified outcome rewards, eventually addressing the sparsified reward and advantage vanishing issues in GRPO. Experimental results on multimodal math and understanding tasks show the effectiveness of Share-GRPO over GRPO.

**Questions:**

See also weakness section for full details.
Q1. In principle, these transformations should preserve task difficulty, so it is unclear why reward signals vary across variants. Can you add more deeper analyses (e.g., probing activation patterns or qualitative error cases) to strengthen the claim. This will impact my final scores significantly.

Q2. Can you also report the full experiment results with Qwen 2.5-VL + GRPO in Table1? This is very important to justify the effectiveness of the proposed Share-GRPO.

**Ethical Concerns:**

["NO or VERY MINOR ethics concerns only"]

**Final Justification:**

Share-GRPO is an interesting work as it mitigates the sparse reward issue by paraphrasing input questions and elicit different rewards for question variants. The results are positive and are better than GRPO.

The origin paper doesn't include direct baseline to justify improvements but the authors provided during rebuttal and hence I am willing to raise my score to 4.

The biggest limitations of this paper to me are
(1) lack of in-depth discussion of the transformation techniques and their impacts on reasoning:
The operations authors selected to transform the input questions are not well motivated. ShareGRPO is using this transformation for reasoning reward diversification while similar techniques were primarily used for data augmentation during training. Although the authors should specify how they verify the transformation results. They include some descriptions in rebuttal but I expect more from a really strong paper.

(2) no improvements between question variants N=2 to 4.
This raises the question of whether the transformation techniques authors selected truly introduce enough variance. How do they verify the transformed inputs and how do they perturb when increasing the variant numbers.
Other reviewers also share the same concern. The author tried to address this by saying the reward spasification issue is mitigated when N=2. But more evidence should be provided.

Overall, this paper is positive but still have some limitations.

**Limitations:**

yes

**Quality:**

3

**Strengths And Weaknesses:**

Strengths
1. Clarity and readability – The manuscript is well-structured and easy to follow, allowing readers to quickly grasp the core ideas and experimental setup.
2. Conceptual simplicity with practical impact – The proposed synthetic-variant expansion is a straightforward yet effective alternative to prior sampling- or selection-based approaches.

Weaknesses & Suggestions:

1. Insufficient analysis of the transformation’s effect on reasoning
The paper lacks a theoretical argument—or at least a more extensive empirical study—showing that the proposed transformations genuinely elicit distinct reasoning behaviors and mitigate GRPO’s sparsified-reward and advantage-vanishing issues.
In principle, these transformations should preserve task difficulty, so it is unclear why reward signals vary across variants. Adding deeper analyses (e.g., probing activation patterns or qualitative error cases) would strengthen the claim.
Table 4 shows nearly identical scores for 2–4 question variants, further calling into question the diversity effect.

2. Expansion-technique details and validation:
Since paraphrasing relies on GPT-4o prompts, a validation step is needed to guarantee semantic equivalence; the limitation is acknowledged but not addressed experimentally.
The chosen image transformations are only listed, not motivated. Please justify each operation (e.g., rotation, noise) and discuss how it differs from standard data-augmentation baselines.

3. Baseline coverage:
A direct comparison between vanilla GRPO and Share-GRPO on the identical backbone (e.g., Qwen 2.5-VL + GRPO vs. Qwen 2.5-VL + Share-GRPO) is essential and should appear in Table 1.
Tables 2–3 cover only a subset of benchmarks. Report GRPO results on all datasets to provide a complete effectiveness picture.

---

> ### Author Rebuttal · Authors · 2025-07-31
>
> Dear reviewer Amky,
>
> We appreciate the reviewer's professional, detailed, and valuable comments and suggestions.
> We sincerely thank you for recognizing our contributions regarding the clarity and readability of the manuscript, the conceptual simplicity and practical impact of our approach, and the effectiveness of Share-GRPO over GRPO.
> Below, we are going to respond to your concerns one by one.
>
> ---
>
> **W1: Analysis of the transformation’s effect on reasoning**
>
> For textual transformations, the overall difficulty of the problem does not change significantly.
> However, it is important to highlight that, based on our experiments and supported by previous studies [1,2], current MLLMs have data leakage and memorization issues.
> Specifically, MLLMs may memorize the correct answers, which allows them to produce the correct final answer even when their intermediate reasoning steps are incorrect, leading to reward hacking issues in GRPO (i.e., cases where models answer all correctly in rollout without genuine reasoning).
> Therefore, by applying textual transformations to introduce new question variants, ShareGRPO can alleviate the sparse reward issue caused by memorization, as responses are generated for the questions the model has not memorized.
> This enables MLLMs to better learn to reason during the ShareGRPO training.
>
> Additionally, MLLMs are sensitive to prompts [3,4]. Even semantically equivalent questions with different formulations can elicit different responses from the model.
> We leverage this property by generating semantically consistent question variants, which encourages more diverse reasoning responses.
> These diverse responses are then shared and used to update all question variants.
> This diversity (which is more likely to include both positive and negative samples) helps to address the advantage vanishing problem.
> At the same time, training with ShareGRPO enhances the model’s robustness to prompt variations.
>
> For multi-modal image transformation, we believe that adding noise or rotating the image increases the difficulty of the problem. This helps reduce the proportion of all-correct cases during training and enhances the model’s robustness to visual variations.
>
>
> > Table 4 shows nearly identical scores for 2–4 question variants, further calling into question the diversity effect.
>
> Based on Table 3 and Table 4, we observe that increasing the number of questions from 1 to 2 raises the accuracy from 72.8% to 75.4%, demonstrating the effectiveness of ShareGRPO compared to GRPO.
> The improvement is limited when increasing from 2 to 4 questions, as with two variants, the MLLM is already able to generate both positive and negative rollouts, effectively mitigating the sparse reward issue. As a result, further gains are less pronounced. Therefore, to balance efficiency and performance, we use two question variants as the default setting.
> In summary, the significant performance improvement when increasing the number of question variants from 1 to 2 demonstrates that our method effectively mitigates the advantage vanishing problem, highlighting the effectiveness of ShareGRPO.
>
> [1] Reasoning or Memorization? Unreliable Results of Reinforcement Learning Due to Data Contamination
>
> [2] MMReason: An Open-Ended Multi-Modal Multi-Step Reasoning Benchmark for MLLMs Toward AGI
>
> [3] How Susceptible are LLMs to Influence in Prompts?
>
> [4] Large Language Models Sensitivity to The Order of Options in
> Multiple-Choice Questions
>
> ---
>
> **Q1: Qualitative error cases about transformations**
>
> This is a good question.
> We agree that providing deeper analyses of how question transformations affect reward signals and model reasoning would strengthen the claim.
> We would like to kindly clarify that we provide some qualitative cases in Appendix E. Due to the technical restrictions from NeurIPS, we used existing images rather than new ones to provide qualitative explanations.
> For Rollout Case No.1-1 in Appendix E, it can be observed that the original question is answered incorrectly. If MLLMs consistently fails on this question, the advantages in GRPO vanish, preventing effective learning.
> However, through semantic-consistent transformations in ShareGRPO, new variants such as No.1-2 and No.1-3 yield correct reasoning paths. These successful trajectories can then be shared back with the original question, allowing the model to recover meaningful optimization signals.
> For Rollout Case No.2-1, it can be observed that the original question is answered correctly. Assuming all responses to its rewritten variants are also correct, GRPO will encounter only successful trajectories, leading to sparse rewards and advantage vanishing.
> By introducing input transformations through ShareGRPO, variants of Case No.2-1 (i.e., No.2-2 and No.2-3) result in incorrect answers. This facilitates learning from both correct and incorrect reasoning trajectories, enhancing the model’s generalization and robustness to question shifts.
>
> We hope these analyses and explanations could address your concerns. We will include them in the revised manuscript. Thank you!
>
> ---
>
> **W2: Expansion-technique details and validation of transformations**
>
> > Since paraphrasing relies on GPT-4o prompts, a validation step is needed to guarantee semantic equivalence;
>
> Thank you for raising this important concern.
> In fact, we conducted extensive experiments to validate the quality of the generated questions.
> 1) We experimented with various prompts to ensure stable and high-quality outputs.
> 2) We developed rule-based scripts to automatically filter out incomplete generations, ensuring that the retained sentences are complete and that the options remain consistent with the original questions.
> 3) We manually reviewed a large number of cases to further ensure the quality and correctness of the outputs.
> 4) We also conducted GRPO training using only one newly generated question variant per sample. The training process was stable and achieved a slight performance improvement as shown below, which empirically demonstrates the quality of the generated questions.
>
> | Models    | one original question | one new question  |
> |-----------|-----------|--------|
> | GRPO      | 72.8      | 72.9   |
>
> > Please justify each operation (e.g., rotation, noise)
>
> we carefully select transformations (e.g., rotation, noise injection) that preserve critical visual cues necessary for reasoning, and avoid transformations (e.g., cropping, color distortion) that may disrupt key information.
> First, we intentionally avoided transformations that could affect critical information in the images. For example, cropping might remove a key edge in a math problem, and color transformations could impact questions that rely on color-related information.
> Next, we selected transformations such as rotation and noise injection, which preserve the critical visual cues necessary for reasoning. Together with our transformation-specific prompts, these changes do not affect the correctness of the problem’s answer.
>
> > discuss how it differs from standard data-augmentation baselines.
>
> I believe there is some misunderstanding here.
> The innovation of this paper does not lie in introducing data augmentation. The main contribution is using carefully designed semantic transformations to generate shared and diverse responses, and then self/cross-updating the question variants with these shared responses.
> Specifically, GRPO calculates the conditional probability of each generated response with respect to the original question and updates the token probabilities accordingly, i.e., $\pi_{\theta}(o_i^{Q} \mid Q)$. The GRPO + data augmentation approach works similarly, except it replaces the original question with the augmented one.
> In contrast, ShareGRPO generates a set of shared responses for the original question and its semantically consistent variants. These responses are then used for both self-updating ($\pi_{\theta}(o_i^{Q_j} \mid Q_k)$, where $j = k$) and cross-updating ($\pi_{\theta}(o_i^{Q_j} \mid Q_k)$, where $j \neq k$), based on their respective conditional probabilities.
>
> Additionally, we compare the results of ShareGRPO and GRPO with data augmentation, as shown below. The results indicate that the performance gains of ShareGRPO are not due to data augmentation, but rather from mitigating advantage vanishing through shared responses.
>
> | Methods    | GRPO | GRPO + image transformations | GRPO + language transformations | GRPO + image/language transformations | ShareGRPO |
> |-----------|-----------|--------|-------|------|------|
> | MathVista | 72.8      | 73.4   | 73.1  | 73.6 | 75.4 |
>
> ---
>
> **W3 & Q2: Compare ShareGRPO with full experiment results with Qwen 2.5-VL + GRPO**
>
> Thank you for your suggestion.
> We have now provided the full experimental results comparing Qwen 2.5-VL + GRPO and Qwen 2.5-VL + ShareGRPO across all benchmarks. The results, shown in Table below, demonstrate that ShareGRPO achieves consistent improvements over GRPO, with an average performance gain of 1.9\% across all benchmarks.
>
> | Models    | MathVista | MMStar | MMMU  | MathVerse | MathVision | AI2D | AVG  |
> |-----------|-----------|--------|-------|------------|-------------|------|------|
> | GRPO      | 72.8      | 65.4   | 56.4  | 50.7       | 26.7        | 84.0 | 59.3 |
> | ShareGRPO | 75.4      | 67.0   | 58.1  | 52.8       | 29.5        | 84.5 | 61.2 |
>
> ---
>
> Many thanks for your professional, detailed, and valuable reviews!
> We have done our best to address each of your concerns and hope our response can resolve them. Please let us know if you have any other questions. If you feel all questions have been addressed, we would greatly appreciate it if you could kindly consider re-rating our work. We are looking forward to hearing from you!

---

> > ### Comment · Reviewer_Amky · 2025-08-04
> >
> > Thanks for the detailed responses! The new numbers you provided are very informative. Especially the complete baseline of full experiment results with Qwen 2.5-VL + GRPO and the results with training on only one new question.
> > I would encourage the authors to include these results in the next revision.

---

> ### Author Response · Authors · 2025-08-05
> **Many thanks for your professional and insightful review!**
>
> Thank you again for your thorough and professional review, which has greatly improved the quality and clarity of our paper.
>
> We will include the complete baseline of all experimental results, the results obtained from training on only one new question, as well as other points discussed in the review, in the revised paper.
> Please feel free to contact us if you have any further questions or suggestions. Thank you!

---

### Decision · Program_Chairs · 2025-09-17

**Decision:**

Accept (poster)

**Comment:**

The paper proposes a new RL method for MLLMs that mitigates the sparsity reward and advantage vanishing problems. This is achieved by generating semantically consistent variants of a given question and then sharing the diverse reasoning trajectories discovered across all variants during policy updates.

There is a consensus to accept the paper after the rebuttal and discussion period. Reviewers recognize the importance of the problem, the novelty of the method, and good presentation and empirical studies.

Overall, this is a solid work to the field of MLLM training. Please incorporate the new results and discussions in the final version.